# Micro-optical elements from optical-quality ZIF-62 hybrid glasses by hot imprinting

Oksana Smirnova[1], Roman Sajzew [1,2], Sarah Jasmin Finkelmeyer [2], Teymur Asadov[1], Sayan Chattopadhyay[1], Torsten Wieduwilt [2], Aaron Reupert[1], Martin Presselt[2,3,4], Alexander Knebel [1,3] & Lothar Wondraczek [1,3] ✉

Hybrid glasses derived from meltable metal-organic frameworks (MOFs) promise to combine the intriguing properties of MOFs with the universal processing ability of glasses. However, the shaping of hybrid glasses in their liquid state – in analogy to conventional glass processing – has been elusive thus far. Here, we present optical-quality glasses derived from the zeolitic imidazole framework ZIF-62 in the form of cm-scale objects. These allow for in-depth studies of optical transparency and refraction across the ultraviolet to near-infrared spectral range. Fundamental viscosity data are reported using a ball penetration technique, and subsequently employed to demonstrate the fabrication of micro-optical devices by thermal imprinting. Using 3D-printed fused silica templates, we show that concave as well as convex lens structures can be obtained at high precision by remelting the glass without trading-off on material quality. This enables multifunctional micro-optical devices combining the gas uptake and permeation ability of MOFs with the optical functionality of glass. As an example, we demonstrate the reversible change of optical refraction upon the incorporation of volatile guest molecules.

Glasses made from meltable metal-organic frameworks (MOFs) and, more specifically, from zeolitic imidazolate frameworks (ZIFs) are about to enter their second decade of discovery[1]. They are being thought of as offering a route for combining the intriguing properties of MOFs with the processing ability of conventional glasses. To date, however, this combination of properties has not yet been demonstrated. MOFs are reticular crystalline materials consisting of inorganic metal nodes interconnected by organic ligands[2]. They feature large porosity, tailorable for a wide range of applications through versatile combinations of node and ligand species[3–5]. Only a few MOFs from the ZIF subclass have been discovered to be meltable before thermal decomposition; subsequent quenching of the melt led to a glassy state (denoted $a_g$) with the ability to retain a certain resemblance to the original MOF[6]. Although some features of ZIF glasses remain disputed[7–9], first reports on the optical[10] and mechanical[11] properties of $a_g$ZIF-62 showed promising results in terms of indeed resembling

traditional glass materials. For the case of cobalt-substituted $a_g$ZIF-62(Co), non-linear optical responses and photoluminescence were reported[12,13]. All of these studies build on the promise that glassy MOFs could − in principle − be processed in their liquid state so as to attain a broad variety of geometrical shapes. This would offer substantial advantages over their crystalline counterparts, which typically require some kind of powder processing and/or binder phases compromising potential applications.

Fabricating dense polycrystalline MOF films with optical quality involves time-consuming, inefficient processing in a highly controlled environment. For example, high-quality, optically transparent HKUST-1 (Hong Kong University of Science and Technology 1) thin films can currently be grown only in an inert, humidity-controlled atmosphere[14]. Some of us demonstrated recently that MIL-68(In) (Matériaux de l'Institut Lavoisier) thin films can indeed be made within minutes[15], but MIL-68(In) currently remains a unique example in those terms.

[1]Friedrich Schiller University Jena, Otto Schott Institute of Materials Research, Fraunhoferstr. 6, Jena, Germany. [2]Leibniz Institute of Photonic Technology (IPHT), Albert-Einstein-Str. 9, Jena, Germany. [3]Friedrich Schiller University Jena, Center for Energy and Environmental Chemistry, Jena, Germany. [4]SciClus GmbH & Co. KG, Moritz-von-Rohr-Str. 1a, Jena, Germany. ✉e-mail: lothar.wondraczek@uni-jena.de

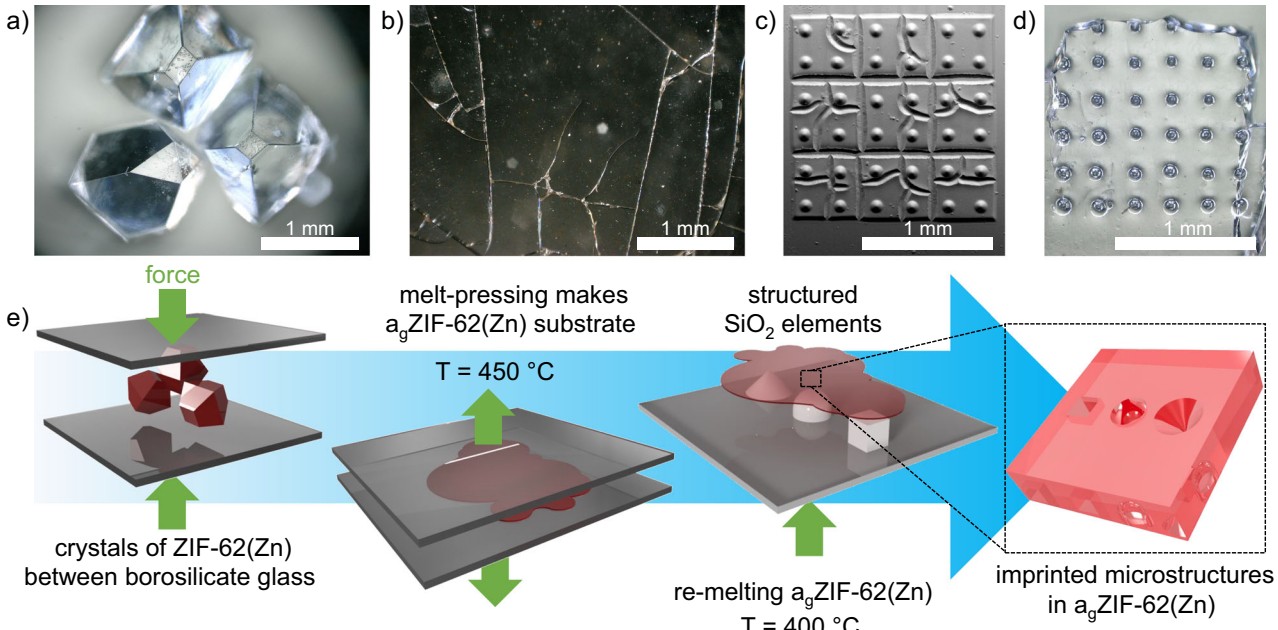

**Fig. 1 | MOF-glass for advanced optical applications: from crystals to responsive micro-optical elements.** Optical micrographs of (**a**) ZIF-62 crystals and (**b**) high optical quality $a_g$ZIF-62 shards, (**c**) exemplary SEM (Scanning Electron Microscopy) image of microstructures printed from GP-Silica resin, (**d**) corresponding optical micrograph of an $a_g$ZIF-62 substrate with imprinted microlenses. **e** Schematic of the workflow from ZIF-62 crystals to imprinted micro-optical components made of $a_g$ZIF-62.

Nevertheless, layers and coatings of MOF materials are highly interesting, as they can combine gas permeation and adsorption performance with optical functions. If processed in the glassy state, not only supported, but also dense, free-standing, transparent films, membranes, and functional substrates could be fabricated. Keppler et al. demonstrated recently that optical ZIF-8 coatings exhibit large shifts in their refractive index depending on the medium they have been exposed to[16]. Similar tuning of the refractive index through the incorporation of volatile guest molecules into the porous channels of optical quality $a_g$ZIF-62 can be envisioned, provided that glass processing can be achieved in the first place.

Thus far, a barrier to progress in the optical utilization of $a_g$ZIF-62(Zn) was the quality of the starting MOF or ZIF precursors and their processed glasses, which led to issues in optical purity, spatial homogeneity, and achievable specimen size[11]. For example, this includes thermal degradation, phase evolution, and gas uptake during extensive re-melting[17,18]. In some cases, MOF melting and glass formation is mediated by auxiliary materials such as ionic liquids; unfortunately, these mediating agents and their decomposition or reaction products again contaminate the final glass, leading to similarly degraded optical quality and, additionally, reduced pore accessibility[19,20]. Recently, we developed a procedure for liquid handling of high-quality ZIF-62-glass with accessible pore channels[9]. This was achieved by optimizing the synthetic conditions of high-quality ZIF-62(Zn) crystals (Fig. 1a) and re-evaluating the melting processes for $a_g$ZIF-62(Zn) (Fig. 1b). The resulting $a_g$ZIF-62(Zn) allows for quantification of gas diffusion coefficients via direct infrared imaging techniques, for which optical transparency is a prerequisite[9].

In this work, we elaborate on the further processing of MOF-glass toward real-world optical applications. For this, we initially revisit the optical properties of $a_g$ZIF-62(Zn) using homogeneous, highly transparent cm-scale samples, which are free of residual contamination, bubbles, cracks, or grain boundaries compromising optical analyses. As a proof of concept for the optical application of $a_g$ZIF-62(Zn), we employ thermal micro-imprinting[21,22] to generate concave and convex micro-optical components (Fig. 1 c–e), on which details of the shaping

process are evaluated. The shaping of the bulk $a_g$ZIF-62(Zn) into such structures provides a perspective for potential use cases of MOF-glasses in photonic devices[23]. In those, for instance, the dependence of the refractive index on the presence of adsorbed molecules can be exploited as a basic principle of function, for example, enabling smart and multi-responsive sensing devices.

## Results and discussion
### Thermal and optical properties of high-quality $a_g$ZIF-62

ZIF-62 is a metal-organic framework with a highly ordered porous structure[24], depicted in Fig. 2a. As a confirmation of the initially crystalline state, powder XRD (PXRD) was performed and compared to the calculated data[25] (Fig. 2b). The XRD pattern of the fully amorphous $a_g$ZIF-62 is also shown in Fig. 2b.

Viscosity data of $a_g$ZIF-62 as presented in Fig. 2c was obtained via a ball penetration technique[26]. For comparison, previous estimates used calorimetric data to extrapolate viscosity using $T_g$ and the liquid fragility index $m$[6]; direct measurements in the supercooled range similar to the present data have not been available. For instrumented penetration experiments, we used a sapphire ball with a radius of $R = 0.75$ mm, loaded with 10-20 mN, on which the rate of sink-in into the glass' surface was determined as a function of temperature (see Methods section for details). At the glass transition temperature $T_g = 322 \,°C$ (determined using differential scanning calorimetry (DSC)[9]; $\log \eta(T_g) = 12$ is assumed for normalizing the viscosity at $T_g/T = 1$, black pentagon in Fig. 2c). The dashed lines are tangents with slope $m$ of the viscosity at $T_g$ of various reference materials[27]. For example, a kinetic fragility index of $m = 16$ indicates a strong glass such as silica[28]. A larger $m$ indicates more pronounced non-Arrhenian behavior. Values of $m \geq 100$ are typical for polymer melts[27]. The experimental data obtained by ball penetration further deviate from the non-Arrhenian prediction, in particular, approaching lower temperature (increasing relaxation time). For $a_g$ZIF-62, we attribute this observation to three possible effects. For one, ongoing MOF pore collapse or mesoscale sintering contributes to a constriction effect which manifests in an apparent decrease in viscosity[9]. Secondly, for

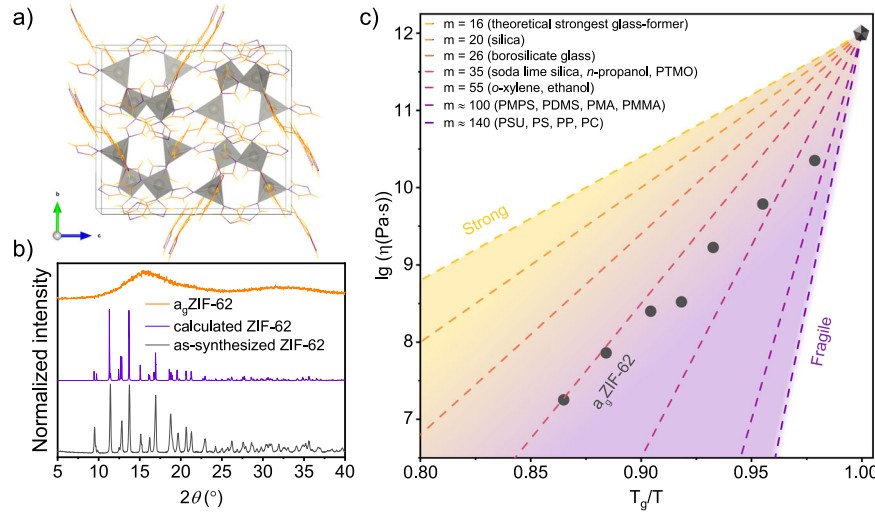

**Fig. 2 | Structure and thermal properties of ZIF-62 and a$_g$ZIF-62. a** Crystal structure of ZIF-62(Zn). Color code: C = orange, N = purple, H = yellow, polyhedra (Zn central atom) = gray. **b** PRXD of ZIF-62 (purple), calculated PRXD of ZIF-62[25] (dark-gray) for comparison and the XRD pattern of the amorphous a$_g$ZIF-62(Zn) (orange). **c** Viscosity of a$_g$ZIF-62 (gray dots) plotted in Angell coordinates (with temperature values converted into kelvins beforehand), and the data of other materials[27] shown for comparison. The dashed lines corresponding to selected values of $m$ are shown to guide the eye. Abbreviations: PTMO = polytetramethylene oxide, PMPS = poly(methyl phenylsiloxane), PDMS = poly(dimethyl siloxane), PMA = poly(methyl acrylate), PMMA = poly(methyl methacrylate), PSU = polysulfone, PS = polystyrene, PP = polypropylene, PC = polycarbonate.

viscosities approaching $10^9$ Pas, the timescale of observation used in the penetration experiment starts to deviate from the timescale of viscous equilibration[29]. Hence, flow is observed in a non-equilibrium state, which leads to the typical deviation between observed (apparent) and stationary viscosity[30]. Finally, an indentation size and strain-rate effect may contribute similarly when approaching higher viscosity (shallow penetration depth)[31]. Linear extrapolation of the data range of $10^7$–$10^9$ Pa·s to $T_g$ gives $m$ (a$_g$ZIF-62) $\approx$ 35, which is in very good agreement with previous evaluations of DSC data[32].

We revisited the optical properties, which were initially published by Qiao et al.[10] on a$_g$ZIF-62 made with a hot-pressing technique. Transmittance spectra of a$_g$ZIF-62 and crystalline ZIF-62 in our work were obtained via three different techniques: direct UV/vis spectrophotometry (Fig. 3a), diffuse UV/vis/NIR transmittance and reflectance spectroscopy (Fig. 3b) and photothermal deflection spectroscopy (PDS, Fig. 3c,d). The direct optical transmittance is strongly dependent on the surface quality and the presence of internal scattering centers (such as pores or grain boundaries) or striae (for example, caused by fluctuations in chemical composition or residual stress). Other than previously reported[10], we find a few distinct shoulders in a$_g$ZIF-62 slightly above the UV absorption edge (Fig. 3a), which we attribute to ligand-to-metal charge transfer[33,34] in accordance with the coordinating nature of bonds within the material, preserved after melting. We assume that this finding is related to material purity and processing parameters; the present a$_g$ZIF-62 did not undergo high-temperature/high-pressure densification upon melting.

Diffuse reflectance spectroscopy (Fig. 3b) allowed us to study also the crystalline ZIF-62 starting material, which was synthesized in powder form. Extending spectroscopy into the NIR region provided additional data for comparing the crystalline precursor and the processed glass. Importantly, almost the full spectral fingerprint of the crystalline ZIF-62 is retained in the processed glass, in particular, without any new vibrations emerging. This is in agreement with previous observations[9] that aside from the release of trapped guest molecules, the relevant ZIF building units (imidazole and benzimidazole linkers) remain intact during glass processing.

Highly sensitive UV-vis-NIR photothermal deflection spectroscopy (PDS)[35–38] was used to investigate the pure absorption (absorptance; free from reflection or scattering losses) of the ZIF-62 materials,

both crystalline and amorphous (Fig. 3 c, d). In PDS, the sample surface is irradiated with modulated monochromatic light in the 200 to 2500 nm range. Absorption leads to local heating, which results in the formation of a thermal lens in the inert liquid (Fluorinert FC-770, Sigma-Aldrich) surrounding the sample. This thermal lens deflects a 633 nm probe-laser guided parallel to the sample surface. This deflection is proportional to the absorption. Unlike the other employed methods, PDS is not limited by the sample thickness, or surface quality (surface roughness is tolerated until thermal lens focusing becomes an issue - fixed crystals on a glass substrate of grain size ~200 μm or less are good to measure, larger crystals become problematic), inner defects and, more important in the present case, the porosity of the material. The PDS absorption data (Fig. 3c) is transformed into an internal transmission spectrum using the Beer-Lambert equation (Fig. 3d). While extrinsic effects such as surface reflection and scattering always affect the observed total transmittance, for the present case, bulk scattering effects are of particular interest, which can be judged from a comparison of PDS and conventional UV/vis/NIR spectroscopic data. Overall, the internal transmittance of a$_g$ZIF-62 reaches 99.48% in the visible range, which is much higher than was observed by conventional transmission spectroscopy. Comparing panels b) and d) of Fig. 3, there appears to be a red shift in the fundamental UV absorption edge, however, this impression is caused by the significantly higher sensitivity of PDS at low absorption. In effect, saturation occurs with respect to the photoinduced heat generation and, therefore, with respect to the focal length of the thermal lens[39–41].

The refractive index of a$_g$ZIF-62 determined by ellipsometry, was found to be $n_D$ (589.3 nm) = 1.5802 (Fig. 3e), which is noticeably higher than previously reported (~1.55 for the same ratio of linkers[10]; noteworthy, the glass used in this previous report was prepared by hot-pressing, which would – assumedly – lead to enhanced densification, too). In addition, we also find a higher Abbe number (lower optical dispersion) of 32.8 (SI) as compared to that previous work (~31)[10] (Fig. 3f); overall, the Abbe number remains on a relatively low level, placing a$_g$ZIF-62 in a group with polymeric materials in terms of optical refraction. The microporosity of the material enables the incorporation of guest molecules; moreover, the glassy nature results in tunable porosity[9]. Aside from affecting gas permeation performance, this

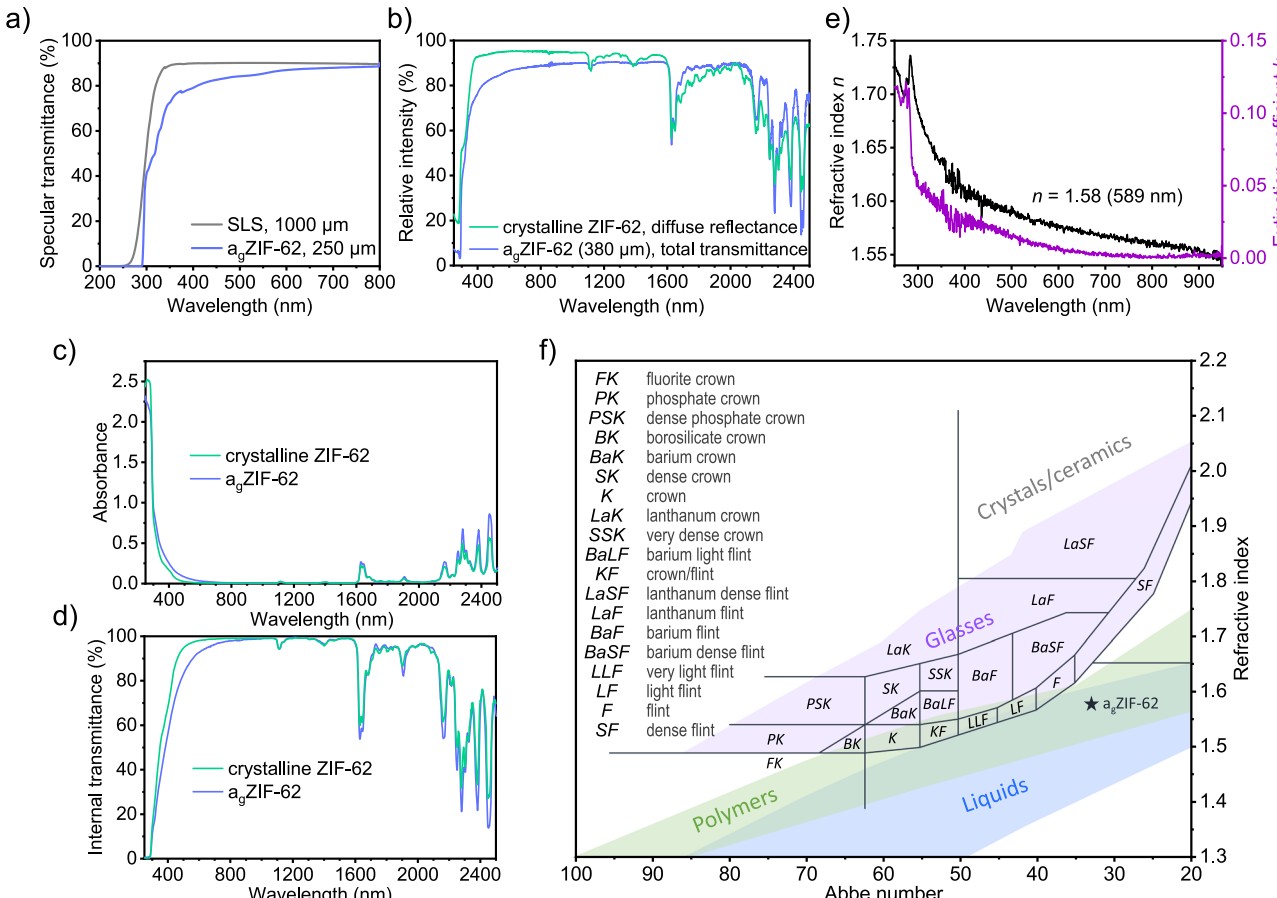

**Fig. 3 | Optical properties of high-quality aₘZIF-62. a** Specular UV/vis transmittance spectra of aₘZIF-62 (blue). Data for soda lime silicate glass (SLS, gray) are shown for comparison. **b** UV/vis/NIR total transmittance and diffuse reflectance spectra of aₘZIF-62 (blue) and ZIF-62 crystals (green), respectively. UV/vis/NIR absorbance (**c**) and internal transmittance (**d**) spectra of ZIF-62 crystals (green) and aₘZIF-62 (blue) obtained by PDS. **e** Refractive index (black) and extinction coefficient (purple) of aₘZIF-62 determined by ellipsometry. **f** Refractive index vs Abbe number ("Abbe plot") adapted from[48] with aₘZIF-62 marked by the dark-blue star. Purple, green, and blue zones indicate glasses, polymers, and liquids on the Abbe diagram, respectively.

would also enable tailoring of the optical properties to an extent beyond the typical cooling-rate dependence of inorganic glasses.

### Thermal imprinting of optical devices on aₘZIF-62

For thermal imprinting, we first generated negative silica templates using a 2PP printing process (see "Methods" section for details). These templates were subsequently used for hot-embossing of target features into the surface of re-heated aₘZIF-62. The overall workflow is shown schematically in Fig. 1e.

Two approaches for thermally imprinting optical structures into the surface of aₘZIF-62 were tested. The first approach included simultaneous melting of ZIF-62 and surface shaping at ~450 °C, in the region of the melting temperature of the crystalline ZIF-62, $T_m$(ZIF-62)[9]. In this case, gently ground crystals were pressed between a glass slide from one side and the 3D-printed template on a substrate from another side. This approach proved insufficient, as the pressure applied to form a single glass piece appeared to be too high for the relatively brittle microstructures, which inevitably led to damaging the template, detaching of its elements, and even glass cracking through induced strain (Figure S1). However, even with this approach, the general ability of 3D imprinting into the aₘZIF-62 surface was clearly observed, as some structures were still thermally transferred (Figure S1). The second approach (Fig. 1e, S1) used a post-processing technique of ZIF-62 glass films prepared beforehand. Here, the template was placed on the top of aₘZIF-62 substrates and then heated up to 400 °C. Gravity and viscous flow led to the imprinting of structures.

Initial tests showed that at this processing temperature, the weight of the substrate itself was sufficient to shape the surface, and no additional external pressure was required for imprinting (Figure S2; this was deemed beneficial for retaining the porosity of the initial aₘZIF-62 also after remelting[9]); also, neither template nor glass was damaged in this process and could be readily detached without leaving residual stress or strain in the material. The second approach was therefore applied in all further experiments.

In Fig. 4 scanning electron microscopy (SEM) images (Fig. 4a,e) and optical profilometry data obtained by laser scanning microscopy (LSM, Fig. 4b,f) of the 3D-printed microstructure are shown. SEM was used for qualitatively inspecting the reproduction of the negative templates on the aₘZIF-62(Zn) surfaces. In the first example, concave template features were used to generate arrays of convex lenses on the aₘZIF-62(Zn) substrate. Digital optical microscopy and LSM images of imprinted features are shown by way of example in Fig. 4c,d, further showing the otherwise unaffected quality of the glass following post-processing. Interestingly, even small imperfections on the 2PP template were reproduced precisely on the aₘZIF-62 substrate, which may be taken as an indicator for the potential performance of thermal imprinting on hybrid glass surfaces.

Noteworthy, there is a difference in depth between the template and imprinted lens (Figure S3). This difference is attributed to delayed elasticity and viscous relaxation; it can both be predicted (so that templates can be designed according to the target imprint geometry) or minimized (through tailoring processing parameters, in particular,

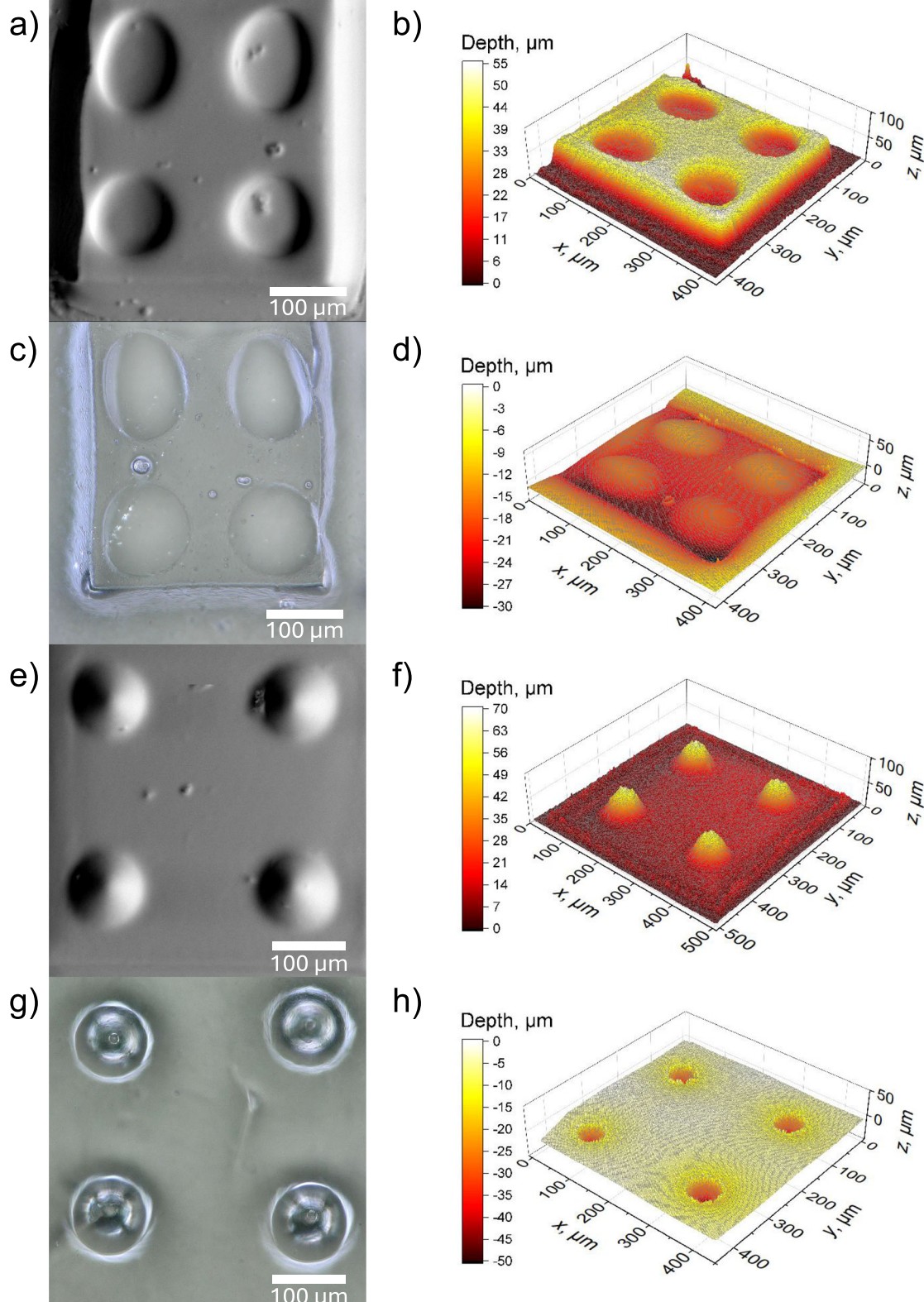

**Fig. 4 | Hot imprinting of optical devices on a$_g$ZIF-62.** Panels **a**–**d** refer to convex lenses. **a** SEM (Scanning Electron Microscopy) and **b** LSM (Laser Scanning Microscopy) images of a negative silica template used for imprinting. **c** Optical microscopy and **d** LSM images of the corresponding imprints generated on a$_g$ZIF-62(Zn).

Panels (e-h) refer to concave lenses. **e** SEM and **f** LSM images of a silica template structure used for imprinting. g) Optical microscopy and h) LSM images of the corresponding imprints on a$_g$ZIF-62.

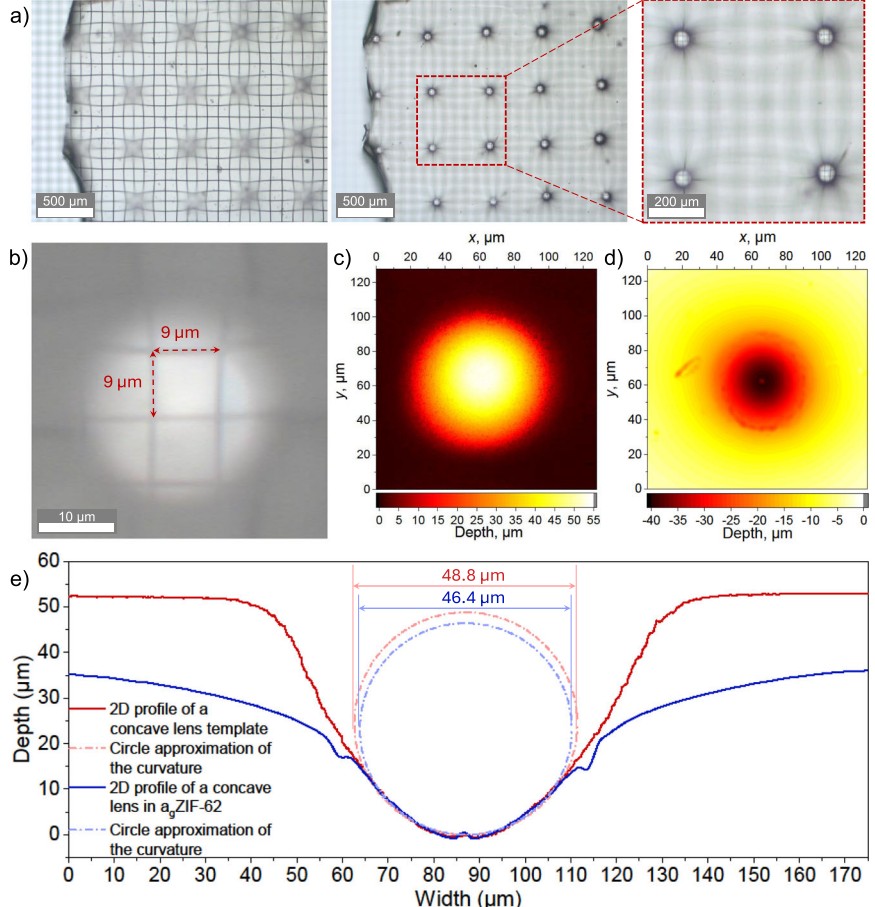

**Fig. 5 | Functioning concave lenses imprinted in $a_gZIF$-62. a** Optical micrographs of $a_gZIF$-62 substrate with imprinted concave lenses placed on the diffraction grating (Carl Zeiss Jena, 20 lines per mm) with the focus on the grating lines visible through the substrate (left) and focal planes of the lenses (middle); zoomed in area (right). **b** Lines of the grating in the focal plane of one of the lenses. 2D LSM (Laser Scanning Microscopy) images of (**c**) one of the lenses on the silica template structure used for imprinting and (**d**) of the corresponding imprint on $a_gZIF$-62 (the same lens is also shown in (**b**)). **e** 2D profiles of (**c**) and (**d**) with their curvatures approximated by the circles obtained from Confomap software.

temperature, time, and normal load). In the latter, a trade-off needs to be considered between retained porosity, and maximum intensity of processing. For the current case, we, therefore, fixed the imprinting temperature at 400 °C (which translates into a viscosity of $10^{7.9}$ Pa·s, right at the limits of practical glass softening) and did not apply any additional normal load (see Fig. 2c).

As another example, we also demonstrate convex imprinting, resulting in concave lenses. The same procedure as with the above convex lenses was applied for this, including the same analytical methods (Fig. 4e–h). There was no major difference in the ease of processing or the quality of attained shapes and retained glass material (Fig. 4e,g).

The concave lenses were studied in more detail in order to demonstrate possible applications of hybrid glass micro-optics. For that, the $a_gZIF$-62 substrate containing imprinted lenses was placed on a diffraction grating with vertical and horizontal lines, forming a defined square mesh with a size of 50 μm (Carl Zeiss Jena, 20 lines per mm) (Fig. 5a). Utilizing an optical microscope, we observed a sharp demagnified grating pattern in the focus of the lenses (Fig. 5 a, b). Compared to the real object size of 50 μm, the diameter of a virtual square in the focal plane of the lens was found to be 9 μm (Fig. 5b), determining the lens magnification of 0.18 in this case. The same lens was then compared to its parent silica template using LSM (Fig. 5 d, e); relatively smooth 2D profiles (Fig. 5f) were obtained from the corresponding LSM images, and their curvatures were fitted with circles using Confomap software. Very close diameters of the fitting circles (48.8 and 46.4 μm) demonstrate that the imprint almost perfectly

reproduced the intended template shape, only lacking the depth, which, as discussed above, can be either predicted or controlled. By applying the lensmaker's equation to the investigated plano-concave lens, the focal length $f = -39.99$ μm was calculated (SI). This value agreed with the experimentally observed focal length, −38.24 μm, which was obtained by multiplying the optical path between the lens and the focal point by the refractive index of $a_gZIF$-62.

The two experiments provide examples demonstrating the fabrication of high-quality $a_gZIF$-62 micro-optical components through thermal imprinting techniques. Without aiming for process optimization at this stage, this proves the ability of MOF-derived glasses to be processed in their liquid state by a standard glass-forming technique. As expected, we run into several effects that make handling of $a_gZIF$-62(Zn) challenging, such as shrinking of the glass melt due to partially collapsing porosity[9], which requires intricate process control in order to achieve large, crack-free objects[6,17,42]. In order to retain the MOF-like porosity, processing must be conducted in the high-viscosity range, that is, at low temperatures and only moderate mechanical loading. In such a case, we demonstrate a time window of 2 h in which thermal forming is possible without material degradation[9]. Another complication in the shaping of $a_gZIF$-62 lies in the hybrid organic-inorganic nature, which makes it essential to handle the melt under an inert atmosphere. For example, even minor traces of oxidizing gases can trigger partial linker decomposition and loss of transparency[9]. We point out the crucial role of tackling these challenges in order to take hybrid glass applications from speculation to reality.

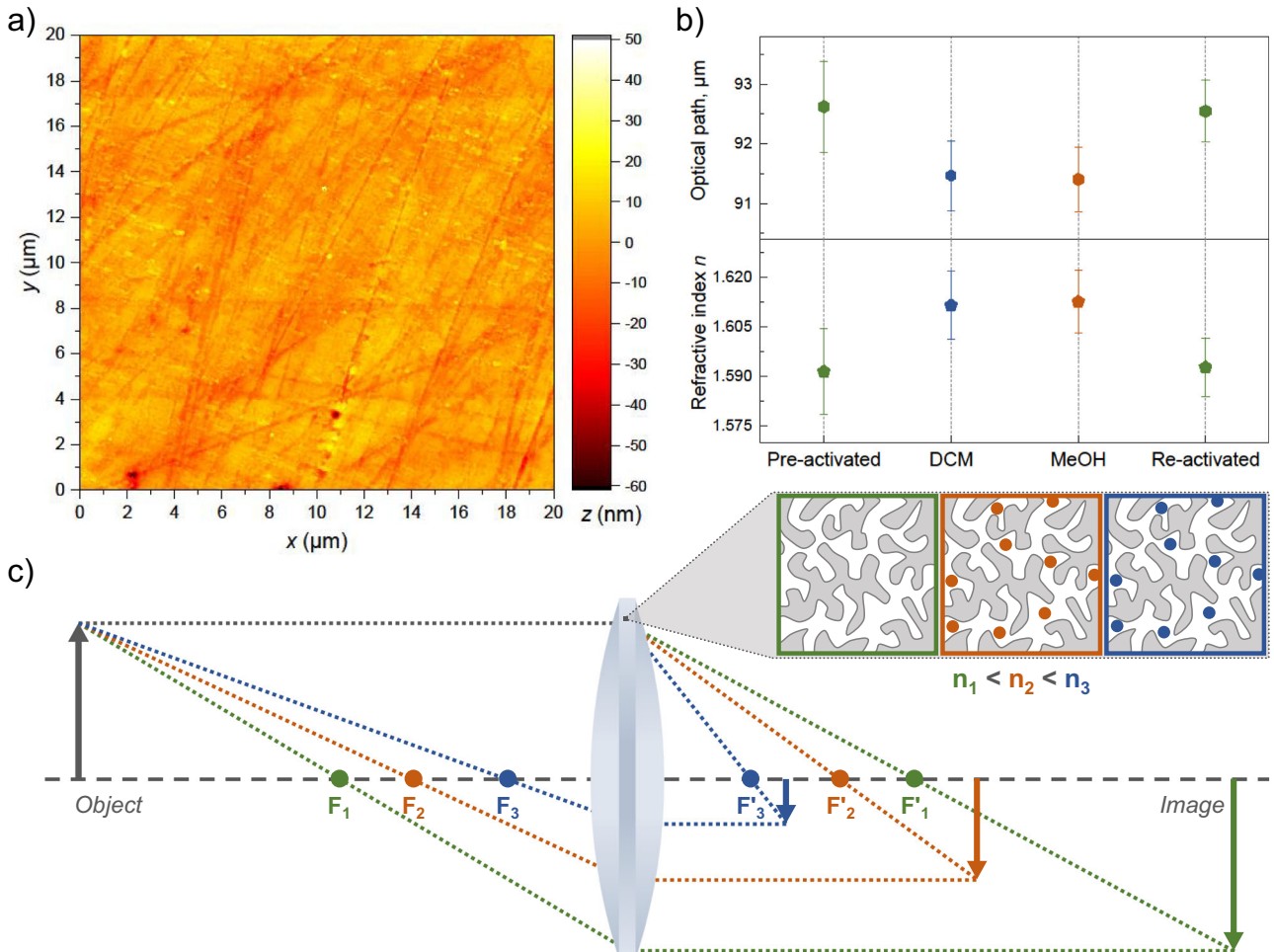

**Fig. 6 | Microporous a$_g$ZIF-62 for responsive optical devices. a** 2D AFM image of the polished surface of a$_g$ZIF-62 for surface roughness determination. **b** Changes in optical path length and refractive index of microporous a$_g$ZIF-62 influenced by different guest molecules. Five profiles of the same sample with 50 μm step were collected in each case. The error bars represent the average absolute deviations of the measured profiles from their concatenate linear approximation. **c** Schematic representation of a perspective responsive micro-optical element based on the microporous a$_g$ZIF-62.

In this way, not only micro-optical structures can be generated, but also other types of thermally imprinted features, e.g., including regular surface patterns, labels, or channels. As a perspective, these features could be designed in such a way that they are sensitive to the presence of volatile guest molecules, which will be discussed below in detail.

**Microporous a$_g$ZIF-62 for responsive micro-optics**
In real-world applications the surface of an optical element usually must be optically smooth, that is, its roughness should be considerably smaller than the wavelength of incident light. Then, surface scattering becomes neglectable, which is especially important in the case of micro-scale optical devices. When a device is manufactured by hot-imprinting or molding, its surface roughness will be strongly affected by the texture of the mold, and using smooth templates is beneficial[43]. Otherwise, the ability of a material to achieve a fine finish by polishing is required. Therefore, despite the well-known softness of MOF-glasses compared to oxide glasses[44], we show that a$_g$ZIF-62 still can be polished in a straightforward way. The surface of a$_g$ZIF-62 pieces was polished with 1 μm diamond polishing spray on a rotating polishing felt, and the resulting surface roughness was determined by atomic force microscopy (AFM) (Fig. 6a). With the maximal deviations staying within the nanometer-range, the average surface roughness Ra was determined to be less than 5 nm. Considering that mechanical polishing of soft materials is not a primary focus of the current report,

achieving such low roughness values in such a facile approach is a strong indicator of the suitability of MOF-glasses for optical polishing.

In Fig. 6b we demonstrate that the introduction of guest molecules into the pores of MOF-glass can be utilized to significantly change the optical properties. The optical path length in a$_g$ZIF-62 was measured with white light after being exposed to the different media, and the respective refractive indices were calculated (see "Methods", Figures S4-S5). First, the optical path length was determined for a pre-activated sample. Then, exposure to dichloromethane (DCM), and subsequently to methanol (MeOH) led to an increase in the refractive index by ~1.26% and ~1.33%, respectively. The initial refractive index (and optical path length) was fully recovered through the re-activation of the sample by heating at low pressure.

To this end, we demonstrate that adsorbed volatile molecules influence the refractive index of the system and that this process is reversible, resulting in optical breathing[45]. Moreover, neither activation nor exposure to the selected solvents damaged the material on the macro-scale, as it remained visually identical between the experiments (Figure S6). This result paves a way for realizing responsive optical elements based on microporous MOF-derived glasses, e.g., stimulus-responsive micro-lenses becoming parts of optical sensing devices (Fig. 6c).

As has been shown before, the properties of ZIF-62 glass are highly dependent on each step of its production – from synthesis of the crystalline material[6,9,46] to the amorphization approach and further

treatment[9,42,47]. Not only porosity, but overall flexible and adaptive microstructure can undergo significant changes due to the high degree of short-range disorder[46], influencing adsorption and optical behavior. Therefore, synthetic and any processing conditions must be thoroughly controlled – and then, in perspective, it will allow for more parameters and possibilities for precise tuning of application-oriented properties.

In summary, hybrid glasses derived from meltable MOFs promise to combine the best of two worlds: the chemical tolerability of MOFs with the processing ability of glasses. We now demonstrated the manufacture of glass objects for micro-optical applications from such hybrid, MOF-derived glasses, by conventional glass forming techniques. For this, we initially revisited the optical properties of $a_g$ZIF-62(Zn), in particular, the UV/Vis/NIR total and internal transmittance, by comparing different techniques and using optical quality materials that have not been available previously. Thereby, it was possible to collect internal transmittance data independent not only from scattering and reflection but also from the porosity and defects of $a_g$ZIF-62(Zn). In this way, we demonstrated optical transmission of up to 99.5% in the visible spectral range, for a sample thickness on the scale of a millimeter. The refractive index and Abbe number were found to be higher than previously reported[10], which is an effect of the improved $a_g$ZIF-62(Zn) purity and overall optical quality. Viscosity data were subsequently collected directly, below $T_m$ ($a_g$ZIF-62(Zn)), using the ball penetration technique. Such data are a prerequisite for hot processing but had previously been available only through extrapolation using the liquid fragility index. In particular, we monitored non-equilibrium viscosity during micro-penetration. Using these data, a process for fabricating micro-optical elements through thermal imprinting techniques was developed. For this, negative templates were prepared by two-photon lithography and used for producing arrays of convex as well as concave micro-lenses on $a_g$ZIF-62(Zn) substrates. The optical performance of such lenses was also reported. This not only demonstrates the first real-world example of hot glass forming using a hybrid, MOF-derived glass and, by this, that the properties of MOFs can indeed be combined with conventional glass processing. We demonstrated that utilizing the porous system remaining in the MOF-glasses, as their unique feature, enables responsive micro-optical devices that shift their optical properties in response to the adsorption and desorption of guest molecules; a new route towards stimulus-responsive optical sensing materials.

## Methods

### Materials
Zinc nitrate hexahydrate (≥ 99%) was purchased from ABCR. Benzimidazole (≥ 99%) was supplied by Alfa Aesar and Imidazole (≥ 99.5%) was purchased from Sigma-Aldrich. N,N-Dimethylformamide (≥ 99.9%) was supplied by VWR, dichloromethane (≥ 99%, stabilized with ethanol) – by Acros Organics. GP-Silica resin and substrates (fused silica and silicon 3D LF DiLL, 25 × 25 × 0.725 mm substrates) were purchased from Nanoscribe GmbH. Methanol (≥ 99.9%) and isopropanol (≥ 99.5%) were supplied by Merck and ROTH, respectively.

### Synthesis of ZIF-62(Zn)
The synthetic procedure of ZIF-62(Zn) was adopted from our previous work[9]. 12.79 g of benzimidazole and 38.02 g of imidazole were successively dissolved in 480 ml of DMF. The solution was stirred for 5 minutes. Subsequently, 19.93 g of Zn nitrate hexahydrate was added to the mixture. The solution was stirred further until complete dissolution. The resulting synthetic Zn:Im:bIm molar ratio was approximately 3:25:5. The solution was then transferred into a 500 ml glass jar and kept for 60 h at 130 °C. White sediment was separated from the mother liquor by centrifugation (10 min, 8.528 g), and thoroughly washed with DMF and DCM, 2 times each. Finally, the obtained ZIF-62 powder was tempered (activated) in a vacuum furnace (25 mbar, 150 °C, 72 h).

### Melting of $a_g$ZIF-62
ZIF-62(Zn) melting was conducted in accordance with our previous report[9]. Crystals were manually ground and the resulting powder was placed between two sheets of silica glass (microscope slides). The glass sheet - ZIF-62(Zn) - sandwich was fixed using two metal clamps, and heated slowly to 450 °C in a nitrogen atmosphere, held at this temperature for 5 min, and cooled back to room temperature. The obtained $a_g$ZIF-62 was finally removed from the silica sheets and stored for further use.

### Optical microscopy
A digital Microscope (VHX-6000, Keyence) with a universal zoom lens (VH-Z100UR and VHX-S650) and a free-angle, xyz observation stage was utilized to collect depth-resolved digital images of sample material. In addition, an Axio Imager Z1m was used in bright-field mode to demonstrate the lens effect of the imprinted devices.

### Laser scanning microscopy
A confocal laser scanning microscope (Carl Zeiss LSM 700) mounted on an Axio Imager Z1m was used to obtain further 3D and 2D images and metric depth profiles of imprints.

### Scanning electron microscopy
A low-vacuum scanning electron microscope (JEOL 6510LV) with a back-scattered electron detector was used to obtain the pictures of 3D-printed silica structures. 30 kV acceleration voltage and 40 Pa pressure were applied.

### Powder X-ray diffraction (PXRD)
A Rigaku MiniFlex diffractometer with Bragg-Brentano geometry was used to obtain PXRD data of crystalline and amorphous samples. The 600 W X-ray generator providing Cu Kα radiation with a wavelength of 1.54059 Å was used, and patterns were taken in the range of 5–50° 2θ with a step size of 0.02°.

### Ball penetration viscometry
The viscous behavior of $a_g$ZIF-62 was investigated through ball penetration viscometry, following the work of Douglas et al.[26]. using a vertical thermomechanical analyzer (Netzsch TMA 402 F1 Hyperion) equipped with an alumina punch and well-defined sapphire spheres as penetration objects. Prior to actual measurements, the temperature at the position of the sample was calibrated through reference measurements of a set of pure metals (≥ 99.99%) with well-known melting points in the relevant temperature range (In, Sn, Bi, Zn, Al).

During actual viscometry, the $a_g$ZIF-62 substrates were placed on a polished alumina plate and the sapphire ball ($R = 0.75$ mm) was positioned on top, being loosely held in place by a support ring to prevent extensive lateral movement of the sphere on the flat sample surface. Samples were heated to different temperatures $T > T_g$ at a rate of 2 K/min, and subjected to isothermal treatment for varying times at these target temperatures. During these treatments, a normal load was applied to the sapphire ball at the beginning of the isothermal segment. The recorded normal displacement of the sapphire ball h over time $t$ was zero-point corrected for any offsets based on accumulated expansion during the heating cycle and the displacement jumps upon application of the desired mechanical load (caused by instrument frame compliance). The such-corrected penetration data were fitted using the relation of Douglas et al.[26]:

$$\eta \, (T = const) = \frac{9}{32} \frac{Pt}{\sqrt{2R}} h^{-\frac{3}{2}} \tag{1}$$

where $P$ is the applied normal load on the penetrating sphere with radius $R$, and $\eta$ is the viscosity of the glass at the temperature during the isothermal segment. As this equation is valid only for $h \ll R$[26], the fitting range was limited to $h < 25\,\mu m$, even if larger penetration depths were reached. For measurements at $T \leq 400\,°C$, a normal load of $P = 20\,mN$ was used during the isothermal segment, while a reduction to $P = 10\,mN$ was necessary for measurements above 400 °C.

## UV-Vis-NIR spectroscopy
Specular (direct, unobstructed) transmittance was characterized by a double-beam spectrophotometer (Cary 5000, Agilent) in air. Diffuse reflectance and total transmittance of crystalline ZIF-62 powder and bulk $a_g$ZIF-62 glass, respectively, were characterized using the same spectrometer equipped with an integration sphere (DRA 2500, Agilent). For diffuse reflection measurements, the powder sample was pressed into a glass sample holder with a cavity of $20 \times 20 \times 0.5\,mm$ and flattened with a microscope glass slide to allow for vertical positioning of the sample as required by the device configuration. A Spectralon white standard was used as 100% diffuse reflectance reference. For total transmission measurements (accounting for directly transmitted and forward scattered light), the $a_g$ZIF-62 glass was placed above the integration sphere opening for the incoming beam.

## Photothermal deflection spectroscopy (PDS)
The photothermal deflection spectroscopy (PDS)[35–38] setup used in this study consisted of a light source (LOT-QD; 1000 W Xe high-pressure lamp and a 300 mm monochromator (LOT-QD MSH 300) optimized for a maximum intensity of 200 nm to 2500 nm). The light was modulated by a chopper (Thorlabs) with a frequency of 5 Hz and focused on a spot size of $2 \times 6\,mm$ on the sample through a $f = 75\,mm$ lens. The intensity of the incident light was monitored using a quartz glass plate as a beam splitter placed between the focusing lens and the sample, and a trans-impedance amplified silicon detector (Thorlabs). The deflection of a 0.5 mW HeNe laser (LINOS, $\lambda = 633\,nm$, HeNe 633-0.8-PO, beam diameter of 0.49 mm) was measured with a lateral effect sensor (Thorlabs PDA90). The deflection and reference signals were read out using two lock-in amplifiers (Stanford Research Systems SRS-830). The whole system was operated using Labview for collecting data and performing PDS signal corrections according to the incident light intensity. A glassy carbon plate (3 mm × 30 mm) was used as a reference sample. ZIF-62 and $a_g$ZIF-62 were fixed by a stripe of parafilm on a quartz glass substrate (30 mm × 5 mm) (see Figure S7), which was then fixed in a quartz glass cuvette (CV10Q3500F, Thorlabs) filled with FC-770 (Sigma Aldrich). The PDS measurement was carried out from 200 nm to 2500 nm in 2.5 nm steps, with the monochromator slits at 2 mm the resolution was 5 nm in the wavelength range 200−1000 nm and 10 nm in the wavelength range 1000−1600 nm.

## Refractive index
The dielectric function of the $a_g$ZIF-62 was determined over the spectra range of 250-950 nm using a variable-angle spectroscopic ellipsometer (Sentech, SE850). One angle below (53°) and two angles above (61° and 65°) the Brewster angle of the glass (~57 °) were chosen as angles of incidence $\theta_i$. The back side was roughened to avoid multiple reflections during the ellipsometric measurement.

The dielectric function that was determined as the so-called *pseudo-dielectric* function $\langle \tilde{\varepsilon} \rangle$ calculated directly from the PSI ($\Psi$) and Delta ($\Delta$) values:

$$\langle \tilde{\varepsilon} \rangle = \sin^2 \theta_i \left[ 1 + \tan^2 \theta_i \left( \frac{1-\rho}{1+\rho} \right)^2 \right] \qquad (2)$$

with the complex polarization ratio

$$\rho = \frac{r_p}{r_s} = e^{i\Delta} \tan \Psi \qquad (3)$$

where $r_p$ is the reflectivity for p-polarized light and $r_s$ is the reflectivity for s-polarized light. In these calculations, it was assumed that the glass sample is an isotropic bulk material, the measuring surface is perfectly smooth without an overlayer and the sample can be considered as an infinite half-space.

## Templates
COMSOL® software was used to create a model of 3D fused silica microstructures. This model was converted into an input file for 3D printing, adjusting hatching, slicing, laser power, and scan speeds using DeScribe software (Nanoscribe GmbH). The object was then printed with the NanoWrite software (Nanoscribe GmbH) on a 2PP lithography system (Photonic Professional GT2, Nanoscribe GmbH). First, GP-silica resin was drop-casted from the cartridge onto a silica substrate. Polymerization was initiated by a pulsed femtosecond fiber laser at 780 nm. During the printing process, laser power and laser scanning speeds were set at 40 mW and 100 mm s⁻¹, respectively. In the development step, the excess resin was removed by keeping the substrates in methanol for 10 min, and in isopropanol for another 1 min. Thermal post-processing (debinding and sintering) was conducted by heating from room temperature to consecutive holding temperatures at rates of 1 °C min⁻¹. Holding temperatures where 90 °C, 150 °C, 230 °C, 280 °C, 600 °C, 1000 °C and 1100 °C, at which the object was held isothermally for 180 min, 180 min, 180 min, 180 min, 120 min, 500 min and 180 min, respectively. Final cooling was done at a rate of 2 °C min⁻¹. This procedure yielded microstructures, attaining 3D resolution in the range of tens of micrometers and surface roughness of 6 nm (Figure S8). It is worth noting that the heat treatment steps affected the structures, and some defects can be observed (e.g., in Fig. 1d). The final printing quality can most probably be improved by adjusting the treatment parameters, which was, however, not in the scope of this study.

## Thermal imprinting
The 3D-printed negative templates were placed on the $a_g$ZIF-62 surface, heated to 400 °C in $N_2$ atmosphere, and kept for 2 h (see main text for further details). After cooling back to room temperature, the silica templates were easily removed from the glass surface to reveal the imprint (Figure S8).

## Atomic force microscopy (AFM)
Prior to the measurement, the sample was glued to a glass slide and wet ground with water using 1000 grit paper and subsequently polished with $1\,\mu m$ diamond polishing spray on a soft polishing felt. The surface topography of the polished glass was characterized using a commercial AFM system (Dimension Edge, Bruker). Measurement was carried out in Tapping Mode, using a silicon tip with a radius of 8 nm and a drive frequency of 209 kHz. Individual measurements were taken at multiple positions on the surface with different sizes ranging from $5 \times 5\,\mu m^2$ to $100 \times 100\,\mu m^2$. Data processing was done using the free Gwyddion software package, v2.65. Post-processing was limited to data leveling and subtraction of a polynomial background of 2nd degree to remove surface waviness, prior to statistical analysis within the software to derive the roughness characteristics.

## Optical path length determination
A transparent piece of agZIF-62 was placed on the diffraction grating (Carl Zeiss Jena, 20 lines per mm). A digital microscope (VHX-6000, Keyence) with a universal zoom lens (VH-Z100UR and VHX-S650) was utilized to obtain 3D images of the edge of $a_g$ZIF-62 piece with

different incorporated guest molecules (Figure S4). First, the sample was pre-activated (150 °C, 20 mbar, 24 hours) to get rid of the possible volatile non-air molecules, and the optical path length was evaluated. Then the microporous glass was left soaking in dichloromethane (DCM) for two hours, dried on the surface, and tested again. After that, the sample was exposed to methanol (MeOH) for 2 hours and re-activated, followed by the optical path length determination after each step. The images were collected in coaxial white light from the bottom of the sample (diffraction grating) to the very sharp image of the grating in the sample without reaching the surface to avoid artifacts. Five 2D profiles of the same sample with 50 μm step were collected for each experiment, processed in an identical way, and approximated by concatenate linear fit (Figure S5). The differences in depths were determined at the edge of the sample, and the value was subtracted from the geometrical sample thickness, resulting in the optical path lengths shown in Fig. 6b of the main text. Refractive indices were calculated by dividing the sample thickness (i.e., geometrical optical path length) by the obtained optical path length. The errors were calculated based on the average absolute deviations of the profiles from their concatenate linear approximation.

## Data availability

All data supporting the findings in this study are available within this Article and its Supplementary Information. Source data are provided in this paper. Any additional data are available from the corresponding author upon request. Source data are provided in this paper.

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

## Acknowledgements

O.S., A.K., and L.W. are supported by the Carl-Zeiss Foundation within the "Breakthroughs 2018" program. This work was further supported by the German Science Foundation (DFG) within the Priority Program SPP 1928/2. We thank Nadja Greiner-Mai (OSIM) for her technical support. S.J.F. and M.P. acknowledge funding from the DFG (Project B09 within the Sonderforschungsbereich SFB/TRR 234 "Catalight", project ID: 364549901). M.P. acknowledges funding from BMWI (ZIM project 217090). We are grateful for the fruitful discussions and support of Dr. Felix Herrmann-Westendorf (IPHT) and Dr. Zhiwen Pan (OSIM). We thank Matthias Arnz (IPHT) for his assistance with polishing the sample and Andrea Dellith (IPHT) for AFM measurements.

## Author contributions

L.W. conceived the idea and managed the project. L.W., A.K., and M.P. provided funding and resources. O.S. synthesized and melted the materials and performed optical micrography, X-ray diffraction, and optical path length/refractive index change determination. R.S. collected viscosity data, made SEM images, and performed specular UV/vis transmittance, UV/vis/NIR total transmittance, and diffuse reflectance measurements. T.A., A.R., and S.C. prepared silica-negative templates. T.A. and O.S. performed thermal imprinting. T.W. collected and analyzed ellipsometry data. A.R. and T.A. performed LSM measurements. A.R. and O.S. tested the functionality and quality of the lenses. S.J.F. and M.P. collected and analyzed UV/vis/NIR absorbance and internal transmittance data by PDS. O.S., A.K., and L.W. analyzed and interpreted the data and wrote and revised the manuscript. All authors added and commented on the manuscript and its revisions.

## Funding

## Competing interests

The authors declare no competing interests.
