## [Peer Review File · Nature Communications]

Micro-optical elements from optical-quality ZIF-62 hybrid glasses by hot imprintingReviewers' comments:

Reviewer #1 (Remarks to the Author):

In the present manuscript, the authors report a ZIF-62-Zn glass. The glass is formed by hot imprinting.

The work lacks novelty for a broad community.

The shaping, casting, forming and hot-pressing of ZIF glasses were presented before and are state of the art. E.g. <https://doi.org/10.1021/acs.chemrev.1c00826>, <https://onlinelibrary.wiley.com/doi/full/10.1002/adfm.202307226> and <https://doi.org/10.1016/j.xcrp.2022.100932>.

So, the presented work contains little novelty.

It shall be published in a specialized journal.

Reviewer #2 (Remarks to the Author):

Oksana et al, prepared ZIF-62 glass in the shape of convex and concave structures by thermal imprinting using 3D fused silica template. However, the novelty and importance are poor. I don't think this draft is suitable for Nature Communications and I recommend the transfer of this manuscript to Scientific reports. More specific comments are as follows:

- ZIF-62 glass is interesting material because of its porosity. However, ZIF-62 glass is not suitable for optical applications because of its poor mechanical properties. For example, the fracture toughness of ZIF-62 glass is 0.1 MPa, which is even lower than that of brittle oxide glasses due to the weak coordinative bonds (Zn-N) (Nat Commun 11, 2593, 2020). The low hardness of ZIF-62 glass prohibits this kind of materials for optical applications.

- The high transparency of ZIF-62 glass is already achieved in other reports (Optics Letters 44, 1623-1625, 2019). Can you retain this high transparency for higher thickness samples (e.g., 3-5 mm thickness)? To really embed ZIF-62 glass in optical applications, you should fabricate highly transparent ZIF-62 glass in different shapes with different thickness.

- The authors claim that they prepared high quality printed ZIF-62 glass, however, in Figure 1c, the SEM image of the printed ZIF-62 glass shows several cracks, reducing the glass quality. Can you prepare ZIF-62 glass in the shape of micro-lens? If the authors really want to prepare ZIF-62 glass in different shapes, other technique should be used rather than the thermal printing. I suggest to use injection molding technology for shaping ZIF-62 glass. The thermal printing technique results in the formation of ZIF-62 glass slide contains some curved structures, while, it is difficult to isolate these structures because of the low hardness of ZIF-62 glass.

- What is the practical application of the printed ZIF-62 glass? Can you control the refractive index of the glass?

- The authors should show whether the printed structures contain bubbles. For optical applications, the surface roughness of optical lens should be less than $Ra < 0.05 \mu\text{m}$. The authors should also measure surface roughness (Ra) for the printed glasses to determine whether it has high quality.

Reviewer #3 (Remarks to the Author):

This is an interesting work. It reports the fabrication of micro-optical elements using MOF glass. Both concave and convex lens structures were successfully produced on the surface of bulk MOF glass through the thermal imprinting method. Subsequently, various optical properties of these micro-optical elements were thoroughly examined. Despite being the first report on micro-optical elements utilizing MOF glass, the authors did not sufficiently describe the significance and novelty of their study. This is important for meeting the standards of Nature Communications. I recommend that the authors address the following issues.

1. The integration of optical functionality and gas permeation in a glass device appears to be

innovative. Nevertheless, it would be valuable to elucidate the practical applications of such devices. Can you provide an example to illustrate their potential utility?

2. In Fig. 3f, it is evident that the optical properties of agZIF-62, specifically the refractive index and Abbe value, are ordinary and comparable to those of polymers. Thus, it does not seem to be an ideal candidate for micro-optical elements. To provide clarity, please elaborate a bit more on the motivation behind this work.

3. This work exclusively presents limited optical properties of the MOF glass microlenses. However, there is no information provided regarding their refractive and diffractive properties. Could you elaborate on these aspects to provide a more comprehensive understanding of the microlenses' optical characteristics?

4. In the abstract section, the assertion that the "shaping of hybrid glasses in their liquid state has been elusive thus far" seems inappropriate. Numerous prior studies have documented the successful preparation of bulk MOF glass. Could you revise this statement to align with the existing literature on the subject?

5. What causes the extensive generation of cracks around the concave or convex lenses, as observed in Fig. 1c, 1d, Fig. 5b, and 5c? Additionally, are there any proposed methods to mitigate the occurrence of these cracks? Addressing this issue is crucial as cracks in the lens can significantly degrade optical performance.

6. Considering the high viscosity of the ZIF-62 melt, is the current melt-pressing approach capable of effectively removing bubbles from the melt? Please describe this aspect in the manuscript.

7. On page 8, in the first paragraph, the authors attribute the shoulder peak at 290-300 nm to the unspecified energy levels of the agZIF-62 glass. Please give additional evidence or discussion to support this assignment.

Reviewer #4 (Remarks to the Author):

This paper reports the fabrication of high-quality optical microlenses from a hybrid glass (ag-ZIF62) using classical glass forming techniques. This is made possible by initially synthesizing ag-ZIF62 a unprecedented material quality (homogeneity, optical transparency). While ag-ZIF62 has become something like the archetype hybrid glass, sample material with similar quality was previously not available. As a great step forward, such high-quality material now enables a great leap also in the quality of property data; in the context of optical devices, UV-NIR spectroscopy, optical refraction and optical dispersion are presented. More compellingly, direct viscosity data are also obtained for the first time by a ball penetration technique, demonstrating equilibrium as well as non-equilibrium viscosity. From such viscosity data, real-world processing windows can be defined within which hot glass forming is possible. Specifically, microlenses are generated by hot imprinting, and subsequently examined for their optical properties. Overall, the paper does not only demonstrate the use of a hybrid glass as a real-world optical material (for example, for use in responsive optical devices). More importantly, it demonstrates for the first time that the intriguing properties of a MOF can indeed be combined with the universal and facile hot glass forming ability; a promise which had previously stipulated intense research in the growing field of MOF glasses, but was never fulfilled until now.

(1) In Figure 3b, there is still some loss at near-infrared waveband region. What is the origin of these absorptions?

(2) The authors demonstrate the potential inactive optical applications. Can this new glass can host active dopants and has potentials for active photonics? For examples, as gain materials for optical amplifier and laser (Adv. Sci. 2023, 10, 2303421; Adv. Opt. Mater., 2021, 9, 2101394.)?

(3) How about the optical quality of the convex lenses in Figure 4?

We would like to thank the four expert reviewers for taking the time to consider our work in detail. The current revision was done in answer to the comments and suggestions provided by the reviewers as well as during the appeal process. Below, is a step-by-step response as to how the manuscript was amended in answer to each comment.

Reviewer #1 (Remarks to the Author):

In the present manuscript, the authors report a ZIF-62-Zn glass. The glass is formed by hot imprinting.

The work lacks novelty for a broad community.

The shaping, casting, forming and hot-pressing of ZIF glasses were been presented before and are state of the art.

E.g. <https://doi.org/10.1021/acs.chemrev.1c00826>, <https://onlinelibrary.wiley.com/doi/full/10.1002/adfm.202307226> and <https://doi.org/10.1016/j.xcrp.2022.100932>.

So, the presented work contains little novelty.

It shall be published in a specialized journal.

We thank the Reviewer for their time and consideration, but must respectfully disagree with the statements, emphasizing the lack of novelty in our work. For that, we refer to our arguments which have been already formulated during the appeal process:

From the appeal letter to the editor (09.02.24):

“Carefully reading the reviewer comments, we acknowledge that the highly multidisciplinary nature of our manuscript – and the fact that hybrid glass research, on the contrary, is still mostly done in the field of chemistry – might have made the review process challenging. This might have led to misunderstandings, which we aim to resolve in the following. We do not think, however, that the multidisciplinary readership of NComms would face a similar issue. On the contrary, we are convinced that the current manuscript will make the field of hybrid glasses significantly more accessible for researchers, engineers and technologists outside of the immediate field of materials chemistry in that it showcases the real-world value of hybrid glasses. This could provide great opportunities; it would stipulate intense, application-oriented efforts beyond materials chemistry while, at the same time, being further motivation and justification for extended research across the material sciences.

[...]

MOF-glass objects intentionally shaped through hot glass forming and, in particular, micro-optical devices such as lenses or structured surfaces are not known to us and are not reported – to our best knowledge – in current literature.

Reviewer 1 – in their very short comment – argues that “shaping, casting, forming and hot-pressing” (of MOF-glasses) are state of the art. We respectfully disagree with this statement in that it does not refer to conventional glass processing (the great MOF promise). Of the three papers mentioned by this reviewer to substantiate their claim, we would limit our initial response to the review paper published in Nov. 2023, Adv. Funct. Mater., <https://onlinelibrary.wiley.com/doi/full/10.1002/adfm.202307226> :

Chapter 3.3 of this paper is devoted to “Morphological Control” of MOF/COF glasses. This chapter explicitly mentions that (conventional glass processing) “is hindered by the relatively narrow

temperature window between T_m and T_d " (the melting and the decomposition temperature of the MOF). Instead, the authors argue that alternative approaches are required, and mention vacuum-assisted sintering or re-melting (but noting that the latter typically leads to poor sample quality). Figure 6 provides good examples of the kind of samples currently obtained in this way; we believe that the issues with sample quality are obvious in these photographs. In addition, the review mentions the ability to "mold cast" certain coordination polymers (ZnPlm); a photograph of such a material is provided in Fig. 3C: it is clearly opaque-white and certainly not a typical glass. We argue that not only do we prove that conventional glass processing is indeed possible, the sample quality obtained in our case is indeed a game-changer towards real-world application (a showcase example of this very glass is provided, on another note, through our recent contribution to the cover illustration of *Nat. Mater.*'s February issue¹). Furthermore, we would highlight that any alternative processing technique involving pronounced pressure gradients (such as mentioned in the above review paper, but also used in the past by ourselves) is counter-productive in that it leads to a well-known loss of MOF porosity.

The second paper mentioned by this reviewer (*Cell Reports Physical Sciences* 2022, <https://doi.org/10.1016/j.xcrp.2022.100932>) discusses powder-sintering of MOF glasses, a subject we find unrelated to our work. The sample quality obtained in this latter study is shown in Fig. 1d of this paper; it appears out of question to us that this material does not establish novelty over our current demonstration, both in terms of sample quality and size. In the same paper, the authors also reproduce a photograph of one of our own samples from an earlier study, their Fig. 2c (our *NComms* [10.1038/s41467-021-25970-0](https://doi.org/10.1038/s41467-021-25970-0)). But this material, again, is not only deeply coloured as a result of being an IL/MOF composite containing decomposition products, it is also in no way a demonstration of any glass forming (shaping) technology.

[...]

Given the above, we assume that Reviewers 1 and 2 might have misunderstood our study in terms of its technological focus (but rather, they might have understood it as being focused on the physical aspects of glass formation and the making of a monolith). A demonstration of glass forming ability (in the technological sense) requires shaping in the hot state, without causing the object to degrade chemically, for example, by partial decomposition. We are convinced that we have made this demonstration in our manuscript, but agree over the potential dual meaning of the term "glass forming" (in the physical sense, leading to a glassy state, and in the technological sense, shaping the glass).

[...]

For one, repeating our arguments above, we provide evidence for conventional hot processing of MOF-derived glasses. This confirms the material's promise, which has been around for several years by now, but remained largely unproven. Secondly, being prominently present in *NComms*, the subject field of hybrid glasses has been devoted almost exclusively to fundamental exploration of the glassy state and some of the achievable properties of monoliths or membranes. The demonstration of real-world processing "into something useful" will stipulate new interest across this community and beyond, hopefully at some point letting us harvest the fruits of all what has been achieved so far with this new class of glasses."

We would also like to add that after the thorough revision process, our manuscript now offers another indisputable novelty in the field of hybrid MOF-glasses – the optical breathing effect, based on the influence of the guest molecules, adsorbed by the microporous matrix, on the optical properties – particularly the refractive index. This process is reversible, causes no damage to the material, and is highly promising for real-world applications, i.e. responsive optical elements for sensing devices,

¹ This cover and article report a membrane application, but do not refer to hot glass processing or optical devices, therefore, do not compromise novelty of the current manuscript.

detecting contaminations in the ambient atmosphere. Moreover, we now demonstrate the ability of MOF-glass to be polished to achieve optical smoothness. Here we present the new chapter and corresponding additions to Methods and Supplementary Information:

Manuscript:

2.3. Microporous a_9 ZIF-62 for responsive micro-optics

In real-world applications the surface of an optical element usually must be optically smooth, that is, its roughness should be considerably smaller than the wavelength of incident light. Then, surface scattering becomes neglectable, which is especially important in the case of micro-scale optical devices. When a device is manufactured by hot-imprinting or molding, its surface roughness will be strongly affected by the texture of the mold, and using smooth templates is beneficial.^[48] Otherwise, the ability of a material to achieve a fine finish by polishing is required. Therefore, despite the well-known softness of MOF-glasses compared to oxide glasses^[49], we show that a_9 ZIF-62 still can be polished in a straightforward way. The surface of a_9 ZIF-62 pieces was polished with 1 μm diamond polishing spray on a rotating polishing felt, and the resulting surface roughness was determined by atomic force microscopy (AFM) (Figure 6a). With the maximal deviations staying within the nanometer-range, the average surface roughness R_a was determined to be less than 5 nm. Considering that mechanical polishing of soft materials is not a primary focus of the current report, achieving such low roughness values in such a facile approach is a strong indicator for the suitability of MOF-glasses for optical polishing.

In Figure 6b we demonstrate that the introduction of guests into the pores of MOF-glass can be utilized to significantly change the optical properties. The optical path length in a_9 ZIF-62 was measured with white light after being exposed to the different mediums, and the respective refractive indices were calculated (see Methods, Figures S4-S5). First, the optical path length was determined for pre-activated sample. Then, exposure to dichloromethane (DCM), and subsequently to methanol (MeOH) led to an increase in the refractive index by $\sim 1.26\%$ and $\sim 1.33\%$, respectively. The initial refractive index (and optical path length) was fully recovered through re-activation of the sample by heating at low pressure.

Figure 1 a) 2D AFM image of the polished surface of a_9 ZIF-62 for surface roughness determination. b) Changes in optical path length and refractive index of microporous a_9 ZIF-62 influenced by different guest molecules. The error bars represent the average absolute deviations of the measured profiles from their concatenate linear approximation. c) Schematic representation of a perspective responsive micro-optical element based on the microporous a_9 ZIF-62.

To this end, we demonstrate that adsorbed volatile molecules influence the refractive index of the system, and that this process is reversible. Therefore, the optical breathing effect^[47] can be generated in a_9 ZIF-62 and similar MOF-derived glasses. Moreover, neither activation nor exposure to the selected solvents damaged the material on the macro-scale, as it remained visually identical between the experiments (Figure S6). This result paves a way for realizing responsive optical elements based on microporous MOF-derived glasses, e.g., stimulus-responsive micro-lenses becoming parts of optical sensing devices (Figure 6c).

Methods:

Atomic force microscopy (AFM)

Prior to the measurement, the sample was glued to a glass slide and wet ground with water using 1000 grit paper and subsequently polished with 1 μm diamond polishing spray on a soft polishing felt. Surface topography of the polished glass was characterized using a commercial AFM system (Dimension Edge, Bruker). Measurement was carried out in Tapping Mode, using a silicon tip with a radius of 8 nm and a drive frequency of 209 kHz. Individual measurements were taken at multiple positions on the surface with different sizes ranging from 5 x 5 μm^2 to 100 x 100 μm^2 . Data processing was done using the free Gwyddion software package, v2.65. Post-processing was limited to data leveling and subtraction of a polynomial background of 2nd degree to remove surface waviness, prior to statistical analysis within the software to derive the roughness characteristics.

Optical path length determination

Transparent piece of $agZIF-62$ was placed on the diffraction grating (Carl Zeiss Jena, 20 lines per mm). A digital microscope (VHX-6000, Keyence) with a universal zoom lens (VH-Z100UR and VHX-S650) was utilized to obtain 3D images of the edge of $agZIF-62$ piece with different incorporated guest molecules (Figure S4). First, the sample was pre-activated (150°C , 20 mbar, 24 hours) to get rid of the possible volatile non-air molecules, and the optical path length was evaluated. Then the microporous glass was left soaking in dichloromethane (DCM) for two hours, dried on the surface, and tested again. After that, the sample was exposed to methanol (MeOH) for 2 hours and re-activated, followed by the optical path length determination after each step. The images were collected in coaxial white light from the bottom of the sample (diffraction grating) to the very sharp image of the grating in the sample without reaching the surface to avoid artifacts. Five 2D profiles were collected for each experiment, processed in an identical way, and approximated by concatenate linear fit (Figure S5). The differences in depths were determined at the edge of the sample, and the value was subtracted from the geometrical sample thickness, resulting in the optical path lengths shown in Figure 6b of the main text. Refractive indices were calculated by dividing the sample thickness (i.e. geometrical optical path length) by the obtained optical path length. The errors were calculated based on the average absolute deviations of the profiles from their concatenate linear approximation.

Supplementary Information:

Figure S4

An example of dataset for optical path length determination collected by z-axis scanning using digital microscope: 3D image of the edge of $agZIF-62$ piece formed by z-axis scans stacking (top left), 2D profiles collected to determine the difference in depth (top right), an example of a profile (bottom).

Figure S5

2D profiles used for the optical path length determination in pre-activated, soaked in DCM and MeOH, and re-activated α_9 ZIF-62 of $147.4 \mu\text{m}$ thickness; their corresponding linear approximations and calculated depths.

Figure S6

Optical photographs of the same pre-activated, soaked in DCM and MeOH, and re-activated α_9 ZIF-62 piece. The piece remains unchanged (besides the glass dust on the surface of pre-activated sample, which was washed away in the solvent while soaking).

Reviewer #2 (Remarks to the Author):

Oksana et al, prepared ZIF-62 glass in the shape of convex and concave structures by thermal imprinting using 3D fused silica template. However, the novelty and importance are poor. I don't think this draft is suitable for Nature Communications and I recommend the transfer of this manuscript to Scientific reports.

We thank the referee for the time dedicated to collecting valuable comments, which are to be discussed below. However, as in the case of the Reviewer 1, we believe that the impression of the lack of novelty in our work might have been caused by the multidisciplinary nature of the manuscript, leading to confusion in terms. To support our position, we would like to refer to our granted appeal again:

From the appeal letter to the editor (09.02.24):

“Carefully reading the reviewer comments, we acknowledge that the highly multidisciplinary nature of our manuscript – and the fact that hybrid glass research, on the contrary, is still mostly done in the field of chemistry – might have made the review process challenging. This might have led to misunderstandings, which we aim to resolve in the following. We do not think, however, that the multidisciplinary readership of NComms would face a similar issue. On the contrary, we are convinced that the current manuscript will make the field of hybrid glasses significantly more accessible for researchers, engineers and technologists outside of the immediate field of materials chemistry in that it showcases the real-world value of hybrid glasses. This could provide great opportunities; it would stipulate intense, application-oriented efforts beyond materials chemistry while, at the same time, being further motivation and justification for extended research across the material sciences.

[...]

MOF-glass objects intentionally shaped through hot glass forming and, in particular, micro-optical devices such as lenses or structured surfaces are not known to us and are not reported – to our best knowledge – in current literature.

[...]

Given the above, we assume that Reviewers 1 and 2 might have misunderstood our study in terms of its technological focus (but rather, they might have understood it as being focused on the physical aspects of glass formation and the making of a monolith). A demonstration of glass forming ability (in the technological sense) requires shaping in the hot state, without causing the object to degrade chemically, for example, by partial decomposition. We are convinced that we have made this demonstration in our manuscript, but agree over the potential dual meaning of the term “glass forming” (in the physical sense, leading to a glassy state, and in the technological sense, shaping the glass).

[...]

For one, repeating our arguments above, we provide evidence for conventional hot processing of MOF-derived glasses. This confirms the material’s promise, which has been around for several years by now, but remained largely unproven. Secondly, being prominently present in NComms, the subject field of hybrid glasses has been devoted almost exclusively to fundamental exploration of the glassy state and some of the achievable properties of monoliths or membranes. The demonstration of real-world processing “into something useful” will stipulate new interest across this community and beyond, hopefully at some point letting us harvest the fruits of all what has been achieved so far with this new class of glasses.”

We would also like to add that after the thorough revision process, our manuscript now offers another indisputable novelty in the field of hybrid MOF-glasses – the optical breathing effect, based on the influence of the guest molecules, adsorbed by the microporous matrix, on the optical properties –

particularly the refractive index. This process is reversible, causes no damage to the material, and is highly promising for real-world applications, i.e. responsive optical elements for sensing devices, detecting contaminations in the ambient atmosphere.

More specific comments are as follows:

- ZIF-62 glass is interesting material because of its porosity. However, ZIF-62 glass is not suitable for optical applications because of its poor mechanical properties. For example, the fracture toughness of ZIF-62 glass is 0.1 MPa, which is even lower than that of brittle oxide glasses due to the weak coordinative bonds (Zn-N) (Nat Commun 11, 2593, 2020). The low hardness of ZIF-62 glass prohibits this kind of materials for optical applications.

We thank the Reviewer for this comment, which, among others, has motivated us to demonstrate more convincing proofs of the applicability of ZIF-62 glass in the field of optics. But first, we would like to refer to the appeal:

From the appeal letter to the editor (09.02.24):

“We must respectfully but strongly disagree with this statement, as there is no direct connection between mechanical properties and the ability to be utilized for optical applications. Not only inorganic glasses, but also a wide range of polymers or even liquids are used as optical components. Fracture toughness or hardness, as questioned by the Reviewer, is often not a primary factor in materials selection; although it is worth noting, that for the ellipsometry measurements ZIF-glasses have been successfully mechanically polished without overall damage to the samples. Finally, the Reviewer refers to a value of fracture toughness reported in literature (they surely intend to use the unit “MPa.m^{-1/2}”). The reviewer is surely aware of the sample quality used in the particular study mentioned above, and of the fact that mechanical studies very strongly rely on sample quality.”

In addition, now our work contains the AFM data of a_gZIF-62 sample, which has been manually polished to optical smoothness with relative ease. Once again, this proves that mechanical properties of the material – if properly produced and treated – are sufficient for many needs. Some complications in handling of MOF-glass at its early-development stage, when compared to the conventional glasses, are compensated by what this material offers in contrast, particularly tunable, guest-accessible porosity.

- The high transparency of ZIF-62 glass is already achieved in other reports (Optics Letters 44, 1623-1625, 2019). Can you retain this high transparency for higher thickness samples (e.g., 3-5 mm thickness)? To really embed ZIF-62 glass in optical applications, you should fabricate highly transparent ZIF-62 glass in different shapes with different thickness.

The work mentioned by the Reviewer indeed presents the transparency data for ZIF-62-derived glass and is acknowledged in our manuscript. However, a hot-pressing technique, utilized in the mentioned work, with the pressure of 50 MPa applied to the material inevitably leads to trading-off porosity – one of the main benefits of MOF-glass. We believe that our work, among others, lays the foundation for manufacturing MOF-glass in different shapes, most importantly, without significant losses in quality or porosity. This work, however, is focused on the micro-scale shaping, which has not been demonstrated before.

- The authors claim that they prepared high quality printed ZIF-62 glass, however, in Figure 1c, the SEM image of the printed ZIF-62 glass shows several cracks, reducing the glass quality. Can you prepare ZIF-62 glass in the shape of micro-lens? If the authors really want to prepare ZIF-62 glass in different shapes, other technique should be used rather than the thermal printing. I suggest to use injection molding technology for shaping ZIF-62 glass. The thermal printing technique results in the formation of ZIF-62 glass slide contains some curved structures, while, it is difficult to isolate these structures because of the low hardness of ZIF-62 glass.

We notice that the Reviewer unintentionally mixed up the figures and their captions, which has led to confusion. This issue has already been discussed in detail:

From the appeal letter to the editor (09.02.24):

“Figure 1c is a SEM image of the template (not the glass!), which was used for thermal imprinting; a part of the same template is also shown in Figure 4e. The actual glass (with the imprints) is in Figure 1d, and it is crack-free. Although the cracking problem is obvious for the templates (produced by 2-Photon-Lithography using a silica precursor), this is not in the scope of the current study. Instead, we demonstrate that even small defects (on the template) can be transferred to the glass substrate by thermal imprinting, indicating the very high possible spatial resolution of this technique.

Furthermore, we find that this Reviewer is in error suggesting the use of injection molding as an alternative hot glass forming technique. Any technical issues notwithstanding, we argue in the manuscript that processing involving high pressures would trade-off MOF porosity – the very reason why using MOF-glasses in the first place.”

- What is the practical application of the printed ZIF-62 glass? Can you control the refractive index of the glass?

We would like to thank the Reviewer for this comment, as it inspired us to test the optical breathing effect in our material – an idea that we introduced as a perspective in our original manuscript but did not risk approaching beforehand due to the presumptive complexity. We realize that additional data would help the readers of our paper to understand this unique functionality of hybrid, MOF-derived glasses in a more instructive way. Therefore, Figure 6b and 6c demonstrate such new data, along with the following new paragraphs, additions to the Methods section, and Supplementary Information:

Manuscript:

2.3. Microporous a_0 ZIF-62 for responsive micro-optics

[...]

In Figure 6b we demonstrate that the introduction of guests into the pores of MOF-glass can be utilized to significantly change the optical properties. The optical path length in a_0 ZIF-62 was measured with white light after being exposed to the different mediums, and the respective refractive indices were calculated (see Methods, Figures S4-S5). First, the optical path length was determined for pre-activated sample. Then, exposure to dichloromethane (DCM), and subsequently to methanol (MeOH) led to an increase in the refractive index by ~1.26% and ~1.33%, respectively. The initial refractive index (and optical path length) was fully recovered through re-activation of the sample by heating at low pressure.

Figure 2 a) 2D AFM image of the polished surface of a_g ZIF-62 for surface roughness determination. b) Changes in optical path length and refractive index of microporous a_g ZIF-62 influenced by different guest molecules. The error bars represent the average absolute deviations of the measured profiles from their concatenate linear approximation. c) Schematic representation of a perspective responsive micro-optical element based on the microporous a_g ZIF-62.

To this end, we demonstrate that adsorbed volatile molecules influence the refractive index of the system, and that this process is reversible. Therefore, the optical breathing effect^[47] can be generated in a_g ZIF-62 and similar MOF-derived glasses. Moreover, neither activation nor exposure to the selected solvents damaged the material on the macro-scale, as it remained visually identical between the experiments (Figure S6). This result paves a way for realizing responsive optical elements based on microporous MOF-derived glasses, e.g., stimulus-responsive micro-lenses becoming parts of optical sensing devices (Figure 6c).

Methods:

Optical path length determination

Transparent piece of a_g ZIF-62 was placed on the diffraction grating (Carl Zeiss Jena, 20 lines per mm). A digital microscope (VHX-6000, Keyence) with a universal zoom lens (VH-Z100UR and VHX-S650) was utilized to obtain 3D images of the edge of a_g ZIF-62 piece with different incorporated guest molecules (Figure S4). First, the sample was pre-activated (150°C, 20 mbar, 24 hours) to get rid of the possible volatile non-air molecules, and the optical path length was evaluated. Then the microporous glass was left soaking in dichloromethane (DCM) for two hours, dried on the surface, and tested again. After that, the sample was exposed to methanol (MeOH) for 2 hours and re-activated, followed by the optical path length determination after each step. The images were collected in coaxial white light from the bottom of the sample (diffraction grating) to the very sharp image of the grating in the sample without reaching the surface to avoid artifacts. Five 2D profiles were collected for each experiment, processed in an identical way, and approximated by concatenate linear fit (Figure S5). The differences in depths were determined at the edge of the sample, and the value was subtracted from the geometrical sample thickness, resulting in the optical path lengths shown in Figure 6b of the main text. Refractive indices were calculated by dividing the sample thickness (i.e. geometrical optical path length) by the obtained

optical path length. The errors were calculated based on the average absolute deviations of the profiles from their concatenate linear approximation.

Supplementary Information:

Figure S4

An example of dataset for optical path length determination collected by z-axis scanning using digital microscope: 3D image of the edge of α -ZIF-62 piece formed by z-axis scans stacking (top left), 2D profiles collected to determine the difference in depth (top right), an example of a profile (bottom).

Figure S5

2D profiles used for the optical path length determination in pre-activated, soaked in DCM and MeOH, and re-activated α_9 ZIF-62 of $147.4 \mu\text{m}$ thickness; their corresponding linear approximations and calculated depths.

Figure S6

Optical photographs of the same pre-activated, soaked in DCM and MeOH, and re-activated α_9 ZIF-62 piece. The piece remains unchanged (besides the glass dust on the surface of pre-activated sample, which was washed away in the solvent while soaking).

- The authors should show whether the printed structures contain bubbles. For optical applications, the surface roughness of optical lens should be less than $Ra < 0.05 \mu\text{m}$. The authors should also measure surface roughness (Ra) for the printed glasses to determine whether it has high quality.

Glasses, presented in this work, have been prepared based on our earlier developed procedure (Nature Materials, 23, 262-270 (2024)), which leads to the material being defect- and bubble-free. The absence of bubbles can be observed within this work in Figures 4c,g, Figure 5a, Figure S6. Then, we would like to acknowledge the suggestion of the Reviewer and present our valuable addition to the manuscript – surface roughness of as-polished MOF-glass evaluated through AFM. Thus, Figure 6a and new paragraph, as well as some additions to Methods:

Manuscript:

2.3. Microporous $a_9\text{ZIF-62}$ for responsive micro-optics

In real-world applications the surface of an optical element usually must be optically smooth, that is, its roughness should be considerably smaller than the wavelength of incident light. Then, surface scattering becomes neglectable, which is especially important in the case of micro-scale optical devices. When a device is manufactured by hot-imprinting or molding, its surface roughness will be strongly affected by the texture of the mold, and using smooth templates is beneficial.^[48] Otherwise, the ability of a material to achieve a fine finish by polishing is required. Therefore, despite the well-known softness of MOF-glasses compared to oxide glasses^[49], we show that $a_9\text{ZIF-62}$ still can be polished in a straightforward way. The surface of $a_9\text{ZIF-62}$ pieces was polished with $1 \mu\text{m}$ diamond polishing spray on a rotating polishing felt, and the resulting surface roughness was determined by atomic force microscopy (AFM) (Figure 6a). With the maximal deviations staying within the nanometer-range, the average surface roughness Ra was determined to be less than 5 nm. Considering that mechanical polishing of soft materials is not a primary focus of the current report, achieving such low roughness values in such a facile approach is a strong indicator for the suitability of MOF-glasses for optical polishing.

[...]

Figure 3 a) 2D AFM image of the polished surface of a_9 ZIF-62 for surface roughness determination. b) Changes in optical path length and refractive index of microporous a_9 ZIF-62 influenced by different guest molecules. The error bars represent the average absolute deviations of the measured profiles from their concatenate linear approximation. c) Schematic representation of a perspective responsive micro-optical element based on the microporous a_9 ZIF-62.

Methods:

Atomic force microscopy (AFM)

Prior to the measurement, the sample was glued to a glass slide and wet ground with water using 1000 grit paper and subsequently polished with 1 μm diamond polishing spray on a soft polishing felt. Surface topography of the polished glass was characterized using a commercial AFM system (Dimension Edge, Bruker). Measurement was carried out in Tapping Mode, using a silicon tip with a radius of 8 nm and a drive frequency of 209 kHz. Individual measurements were taken at multiple positions on the surface with different sizes ranging from 5 x 5 μm^2 to 100 x 100 μm^2 . Data processing was done using the free Gwyddion software package, v2.65. Post-processing was limited to data leveling and subtraction of a polynomial background of 2nd degree to remove surface waviness, prior to statistical analysis within the software to derive the roughness characteristics.

Reviewer #3 (Remarks to the Author):

This is an interesting work. It reports the fabrication of micro-optical elements using MOF glass. Both concave and convex lens structures were successfully produced on the surface of bulk MOF glass through the thermal imprinting method. Subsequently, various optical properties of these micro-optical elements were thoroughly examined. Despite being the first report on micro-optical elements utilizing MOF glass, the authors did not sufficiently describe the significance and novelty of their study. This is important for meeting the standards of Nature Communications.

We thank the Reviewer for a positive evaluation of our work. Now, after thorough revision, this manuscript not only emphasizes the novelty and importance in a clearer way, but also contains a new proof of applicability in optics – we demonstrate the optical breathing effect, which will be further addressed below the comment 1. Particularly, aside from discussion section, exciting perspectives of the material in the field of optics have been underlined in abstract and conclusions through following sentences:

Abstract: Unique to hybrid glasses, this enables tailorable and stimulus-responsive optical performance, demonstrated by way of example through the reversible change of optical refraction upon the incorporation of volatile guest molecules.

Conclusion: We demonstrated that utilizing the porous system remaining in the MOF-glasses, as their unique feature, enables responsive micro-optical devices that shift their optical properties in response to the adsorption and desorption of guest molecules; a new route towards stimulus-responsive optical sensing materials.

I recommend that the authors address the following issues.

1. The integration of optical functionality and gas permeation in a glass device appears to be innovative. Nevertheless, it would be valuable to elucidate the practical applications of such devices. Can you provide an example to illustrate their potential utility?

We are grateful to the Reviewer for this comment, and hereby present an applicable dependence of the refractive index on the guest molecules, incorporated into the nanoporous matrix of ZIF-62 glass. Figures 6b, new paragraphs in the main text and Methods, and Supplementary Information Figures have been added (the ability of MOF-glass to be polished to achieve optical smoothness is also now demonstrated on the same Figure):

Manuscript:

2.3. Microporous a_9 ZIF-62 for responsive micro-optics

In real-world applications the surface of an optical element usually must be optically smooth, that is, its roughness should be considerably smaller than the wavelength of incident light. Then, surface scattering becomes neglectable, which is especially important in the case of micro-scale optical devices. When a device is manufactured by hot-imprinting or molding, its surface roughness will be strongly affected by the texture of the mold, and using smooth templates is beneficial.^[48] Otherwise, the ability of a material to achieve a fine finish by polishing is required. Therefore, despite the well-known softness of MOF-glasses compared to oxide glasses^[49], we show that a_9 ZIF-62 still can be polished in a straightforward way. The surface of a_9 ZIF-62 pieces was polished with 1 μ m diamond polishing spray on a rotating polishing felt, and the resulting surface roughness was determined by atomic force microscopy (AFM) (Figure 6a). With the maximal deviations staying within the nanometer-range, the average surface roughness R_a was determined to be less than 5 nm. Considering that mechanical polishing of soft materials is not a primary focus of the current report, achieving such low roughness values in such a facile approach is a strong indicator for the suitability of MOF-glasses for optical polishing.

In Figure 6b we demonstrate that the introduction of guests into the pores of MOF-glass can be utilized to significantly change the optical properties. The optical path length in a_g ZIF-62 was measured with white light after being exposed to the different mediums, and the respective refractive indices were calculated (see Methods, Figures S4-S5). First, the optical path length was determined for pre-activated sample. Then, exposure to dichloromethane (DCM), and subsequently to methanol (MeOH) led to an increase in the refractive index by $\sim 1.26\%$ and $\sim 1.33\%$, respectively. The initial refractive index (and optical path length) was fully recovered through re-activation of the sample by heating at low pressure.

Figure 4 a) 2D AFM image of the polished surface of a_g ZIF-62 for surface roughness determination. b) Changes in optical path length and refractive index of microporous a_g ZIF-62 influenced by different guest molecules. The error bars represent the average absolute deviations of the measured profiles from their concatenate linear approximation. c) Schematic representation of a perspective responsive micro-optical element based on the microporous a_g ZIF-62.

To this end, we demonstrate that adsorbed volatile molecules influence the refractive index of the system, and that this process is reversible. Therefore, the optical breathing effect^[47] can be generated in a_g ZIF-62 and similar MOF-derived glasses. Moreover, neither activation nor exposure to the selected solvents damaged the material on the macro-scale, as it remained visually identical between the experiments (Figure S6). This result paves a way for realizing responsive optical elements based on microporous MOF-derived glasses, e.g., stimulus-responsive micro-lenses becoming parts of optical sensing devices (Figure 6c).

Methods:

Atomic force microscopy (AFM)

Prior to the measurement, the sample was glued to a glass slide and wet ground with water using 1000 grit paper and subsequently polished with 1 μ m diamond polishing spray on a soft polishing felt. Surface topography of the polished glass was characterized using a commercial AFM system (Dimension Edge, Bruker). Measurement was carried out in Tapping Mode, using a silicon tip with a radius of 8 nm and a drive frequency of 209 kHz. Individual measurements were taken at multiple positions on the surface

with different sizes ranging from $5 \times 5 \mu\text{m}^2$ to $100 \times 100 \mu\text{m}^2$. Data processing was done using the free Gwyddion software package, v2.65. Post-processing was limited to data leveling and subtraction of a polynomial background of 2nd degree to remove surface waviness, prior to statistical analysis within the software to derive the roughness characteristics.

Optical path length determination

Transparent piece of agZIF-62 was placed on the diffraction grating (Carl Zeiss Jena, 20 lines per mm). A digital microscope (VHX-6000, Keyence) with a universal zoom lens (VH-Z100UR and VHX-S650) was utilized to obtain 3D images of the edge of agZIF-62 piece with different incorporated guest molecules (Figure S4). First, the sample was pre-activated (150°C , 20 mbar, 24 hours) to get rid of the possible volatile non-air molecules, and the optical path length was evaluated. Then the microporous glass was left soaking in dichloromethane (DCM) for two hours, dried on the surface, and tested again. After that, the sample was exposed to methanol (MeOH) for 2 hours and re-activated, followed by the optical path length determination after each step. The images were collected in coaxial white light from the bottom of the sample (diffraction grating) to the very sharp image of the grating in the sample without reaching the surface to avoid artifacts. Five 2D profiles were collected for each experiment, processed in an identical way, and approximated by concatenate linear fit (Figure S5). The differences in depths were determined at the edge of the sample, and the value was subtracted from the geometrical sample thickness, resulting in the optical path lengths shown in Figure 6b of the main text. Refractive indices were calculated by dividing the sample thickness (i.e. geometrical optical path length) by the obtained optical path length. The errors were calculated based on the average absolute deviations of the profiles from their concatenate linear approximation.

Supplementary Information:

Figure S4

An example of dataset for optical path length determination collected by z-axis scanning using digital microscope: 3D image of the edge of agZIF-62 piece formed by z-axis scans stacking (top left), 2D profiles collected to determine the difference in depth (top right), an example of a profile (bottom).

Figure S5

2D profiles used for the optical path length determination in pre-activated, soaked in DCM and MeOH, and re-activated a_9 ZIF-62 of $147.4 \mu\text{m}$ thickness; their corresponding linear approximations and calculated depths.

Figure S6

Optical photographs of the same pre-activated, soaked in DCM and MeOH, and re-activated a_9 ZIF-62 piece. The piece remains unchanged (besides the glass dust on the surface of pre-activated sample, which was washed away in the solvent while soaking).

2. In Fig. 3f, it is evident that the optical properties of agZIF-62, specifically the refractive index and Abbe value, are ordinary and comparable to those of polymers. Thus, it does not seem to be an ideal candidate for micro-optical elements. To provide clarity, please elaborate a bit more on the motivation behind this work.

Referring to our response to comment 1 and the new data, we believe that now we have proved that tunable and accessible porosity offers exceptional benefits, unique to this new class of glasses.

3. This work exclusively presents limited optical properties of the MOF glass microlenses. However, there is no information provided regarding their refractive and diffractive properties. Could you elaborate on these aspects to provide a more comprehensive understanding of the microlenses' optical characteristics?

In the manuscript, we indeed provide refractive index and optical dispersion data, now complemented with further data demonstrating the variability of refractive index upon the reversible introduction of guest molecules into the glass' pores. In this manuscript, however, we did not aim for diffractive optics; we believe that this goes beyond the scope of a single paper. The imprinted lenses are intended "only" a demonstration of the general ability to conventionally shape the material through the hot-imprinting technique. We found it important to show that this process preserves the high optical quality of the original glass, and that the obtained object can indeed be used as lenses, as proven by demonstrating the magnifying ability. However, a more specialized investigation of the lenses' properties was not in the scope of the current manuscript (being intended for a more general audience) and would distract readers from its central objective.

4. In the abstract section, the assertion that the "shaping of hybrid glasses in their liquid state has been elusive thus far" seems inappropriate. Numerous prior studies have documented the successful preparation of bulk MOF glass. Could you revise this statement to align with the existing literature on the subject?

Here we must emphasize that the multidisciplinary nature of the work might have led to the confusion in terms. Within this manuscript, we refer to "glass shaping" as a technological process, which includes the treatment of a previously manufactured glass in a hot state, and not as a process of bulk glass fabrication through melting. As the dual meaning of "glass shaping/forming" – technological and physical – might lead to further confusion, we now refer to it as "conventional glass shaping", which is only attributed to a technological process. The following sentence was changed from:

However, the shaping of hybrid glasses in their liquid state has been elusive thus far.

To:

However, the shaping of hybrid glasses in their liquid state – in analogy to conventional glass processing – has been elusive thus far.

5. What causes the extensive generation of cracks around the concave or convex lenses, as observed in Fig. 1c, 1d, Fig. 5b, and 5c? Additionally, are there any proposed methods to mitigate the occurrence of these cracks? Addressing this issue is crucial as cracks in the lens can significantly degrade optical performance.

Going one by one, Figure 1c only shows a 3D-printed template used for hot-imprinting, i.e. a negative of the lens. Although the cracking, caused by the not-yet-optimized fabrication process, is obvious, it neither affects the lenses of the template nor the imprinted structures in MOF-glass. Both template's and glass's lenses are finely formed, smooth, and symmetrical, as shown by LSM in Figures 5 c,d,e.

Figure 1d does show ZIF-62 glass with the imprinted structures, but there are no cracks caused by the surface micro-structuring. Aside from the edge of the sample (which was selected in a small size to fit

into the measurement cell), all space between the lenses is smooth, crack- and defect-free, just like the lenses themselves (which is clearly visible in Figures 5 a,b).

6. Considering the high viscosity of the ZIF-62 melt, is the current melt-pressing approach capable of effectively removing bubbles from the melt? Please describe this aspect in the manuscript.

Glasses, presented in this work, have been prepared based on our earlier developed procedure (Nature Materials, 23, 262-270 (2024)), which leads to the material being defect- and bubble-free. Optical microphotographs of a_g ZIF-62 pieces in a high resolution within this manuscript (particularly Figures 4 c,g, Figure 5a, Figure S6) do not show any evidence of the bubbles throughout the glass volume.

7. On page 8, in the first paragraph, the authors attribute the shoulder peak at 290-300 nm to the unspecified energy levels of the ag ZIF-62 glass. Please give additional evidence or discussion to support this assignment.

Previously unspecified mentioned energy levels are now attributed to the ligand-to-metal charge transfer (LMCT), as similar patterns have been observed for MOFs. The corresponding part is now corrected from:

Other than previously reported[10], we find a few distinct shoulders in ag ZIF-62 slightly above the UV absorption edge (Figure 3a), which we attribute to discrete, as-of-yet unspecified energy levels of the hybrid glass. We assume that this finding is related to material purity and processing parameters; the present ag ZIF-62 did not undergo high-temperature/high-pressure densification upon melting. From the peak position and the sharp band at wavelength 290-300 nm, we can exclude that this band comes from a ZnO cluster, as these normally absorb in the regions above 350 nm[35,36].

To:

Other than previously reported^[10], we find a few distinct shoulders in a_g ZIF-62 slightly above the UV absorption edge (Figure 3a), which we attribute to ligand-to-metal charge transfer^[35,36] in accordance with the coordinating nature of bonds within the material, preserved after melting. We assume that this finding is related to material purity and processing parameters; the present ag ZIF-62 did not undergo high-temperature/high-pressure densification upon melting.

Reviewer #4 (Remarks to the Author):

This paper reports the fabrication of high-quality optical microlenses from a hybrid glass (ag-ZIF62) using classical glass forming techniques. This is made possible by initially synthesizing ag-ZIF62 a unprecedented material quality (homogeneity, optical transparency). While ag-ZIF62 has become something like the archetype hybrid glass, sample material with similar quality was previously not available. As a great step forward, such high-quality material now enables a great leap also in the quality of property data; in the context of optical devices, UV-NIR spectroscopy, optical refraction and optical dispersion are presented. More compellingly, direct viscosity data are also obtained for the first time by a ball penetration technique, demonstrating equilibrium as well as non-equilibrium viscosity. From such viscosity data, real-world processing windows can be defined within which hot glass forming is possible. Specifically, microlenses are generated by hot imprinting, and subsequently examined for their optical properties. Overall, the paper does not only demonstrate the use of a hybrid glass as a real-world optical material (for example, for use in responsive optical devices). More importantly, it demonstrates for the first time that the intriguing properties of a MOF can indeed be combined with the universal and facile hot glass forming ability; a promise which had previously stipulated intense research in the growing field of MOF glasses, but was never fulfilled until now.

We are very grateful to the Reviewer for a positive evaluation of the work and for sharing our excitement on the topic. Below we will address all their comments individually.

(1) In Figure 3b, there is still some loss at near-infrared waveband region. What is the origin of these absorptions?

Losses at near-infrared region in Figure 3b (as well as reproduced transmittance pattern in Figure 3d) are attributed to the fingerprint region of the material in the manuscript.

(2) The authors demonstrate the potential inactive optical applications. Can this new glass can host active dopants and has potentials for active photonics? For examples, as gain materials for optical amplifier and laser (Adv. Sci. 2023, 10, 2303421; Adv. Opt. Mater., 2021, 9, 2101394.)?

We agree with the Reviewer that the demonstration of a use-case for our material is indeed important, and their suggestions seem promising and are worth investigating. However, within this work we chose to focus on exploiting the materials' porosity as its main feature, as conventional doping can be performed with other glasses as well (which does not decrease a potential interest in a doped MOF-glass). Now we demonstrate that nanoporous a_gZIF-62 can be utilized in responsive micro-optics. Figures 6b,c, new paragraphs in the main text and Methods, and Supplementary Information Figures have been added (the ability of MOF-glass to be polished to achieve optical smoothness is also now demonstrated on the same Figure):

Manuscript:

2.3. Microporous a_gZIF-62 for responsive micro-optics

In real-world applications the surface of an optical element usually must be optically smooth, that is, its roughness should be considerably smaller than the wavelength of incident light. Then, surface scattering becomes neglectable, which is especially important in the case of micro-scale optical devices. When a device is manufactured by hot-imprinting or molding, its surface roughness will be strongly affected by the texture of the mold, and using smooth templates is beneficial.^[48] Otherwise, the ability of a material to achieve a fine finish by polishing is required. Therefore, despite the well-known softness of MOF-glasses compared to oxide glasses^[49], we show that a_gZIF-62 still can be polished in a straightforward way. The surface of a_gZIF-62 pieces was polished with 1 μm diamond polishing spray on a rotating polishing felt, and the resulting surface roughness was determined by atomic force

microscopy (AFM) (Figure 6a). With the maximal deviations staying within the nanometer-range, the average surface roughness R_a was determined to be less than 5 nm. Considering that mechanical polishing of soft materials is not a primary focus of the current report, achieving such low roughness values in such a facile approach is a strong indicator for the suitability of MOF-glasses for optical polishing.

In Figure 6b we demonstrate that the introduction of guests into the pores of MOF-glass can be utilized to significantly change the optical properties. The optical path length in a_g ZIF-62 was measured with white light after being exposed to the different mediums, and the respective refractive indices were calculated (see Methods, Figures S4-S5). First, the optical path length was determined for pre-activated sample. Then, exposure to dichloromethane (DCM), and subsequently to methanol (MeOH) led to an increase in the refractive index by $\sim 1.26\%$ and $\sim 1.33\%$, respectively. The initial refractive index (and optical path length) was fully recovered through re-activation of the sample by heating at low pressure.

Figure 5 a) 2D AFM image of the polished surface of a_g ZIF-62 for surface roughness determination. b) Changes in optical path length and refractive index of microporous a_g ZIF-62 influenced by different guest molecules. The error bars represent the average absolute deviations of the measured profiles from their concatenate linear approximation. c) Schematic representation of a perspective responsive micro-optical element based on the microporous a_g ZIF-62.

To this end, we demonstrate that adsorbed volatile molecules influence the refractive index of the system, and that this process is reversible. Therefore, the optical breathing effect^[47] can be generated in a_g ZIF-62 and similar MOF-derived glasses. Moreover, neither activation nor exposure to the selected solvents damaged the material on the macro-scale, as it remained visually identical between the experiments (Figure S6). This result paves a way for realizing responsive optical elements based on microporous MOF-derived glasses, e.g., stimulus-responsive micro-lenses becoming parts of optical sensing devices (Figure 6c).

Methods:

Atomic force microscopy (AFM)

Prior to the measurement, the sample was glued to a glass slide and wet ground with water using 1000 grit paper and subsequently polished with 1 μm diamond polishing spray on a soft polishing felt. Surface topography of the polished glass was characterized using a commercial AFM system (Dimension Edge, Bruker). Measurement was carried out in Tapping Mode, using a silicon tip with a radius of 8 nm and a drive frequency of 209 kHz. Individual measurements were taken at multiple positions on the surface with different sizes ranging from 5 x 5 μm^2 to 100 x 100 μm^2 . Data processing was done using the free Gwyddion software package, v2.65. Post-processing was limited to data leveling and subtraction of a polynomial background of 2nd degree to remove surface waviness, prior to statistical analysis within the software to derive the roughness characteristics.

Optical path length determination

Transparent piece of agZIF-62 was placed on the diffraction grating (Carl Zeiss Jena, 20 lines per mm). A digital microscope (VHX-6000, Keyence) with a universal zoom lens (VH-Z100UR and VHX-S650) was utilized to obtain 3D images of the edge of agZIF-62 piece with different incorporated guest molecules (Figure S4). First, the sample was pre-activated (150°C, 20 mbar, 24 hours) to get rid of the possible volatile non-air molecules, and the optical path length was evaluated. Then the microporous glass was left soaking in dichloromethane (DCM) for two hours, dried on the surface, and tested again. After that, the sample was exposed to methanol (MeOH) for 2 hours and re-activated, followed by the optical path length determination after each step. The images were collected in coaxial white light from the bottom of the sample (diffraction grating) to the very sharp image of the grating in the sample without reaching the surface to avoid artifacts. Five 2D profiles were collected for each experiment, processed in an identical way, and approximated by concatenate linear fit (Figure S5). The differences in depths were determined at the edge of the sample, and the value was subtracted from the geometrical sample thickness, resulting in the optical path lengths shown in Figure 6b of the main text. Refractive indices were calculated by dividing the sample thickness (i.e. geometrical optical path length) by the obtained optical path length. The errors were calculated based on the average absolute deviations of the profiles from their concatenate linear approximation.

Supplementary Information:

Figure S4

An example of dataset for optical path length determination collected by z-axis scanning using digital microscope: 3D image of the edge of α_0 ZIF-62 piece formed by z-axis scans stacking (top left), 2D profiles collected to determine the difference in depth (top right), an example of a profile (bottom).

Figure S5

2D profiles used for the optical path length determination in pre-activated, soaked in DCM and MeOH, and re-activated α_0 ZIF-62 of $147.4 \mu\text{m}$ thickness; their corresponding linear approximations and calculated depths.

Figure S6

Optical photographs of the same pre-activated, soaked in DCM and MeOH, and re-activated α -ZIF-62 piece. The piece remains unchanged (besides the glass dust on the surface of pre-activated sample, which was washed away in the solvent while soaking).

(3) How about the optical quality of the convex lenses in Figure 4?

Unfortunately, we could not detect the magnifying ability of the convex lenses. This can be attributed to the asymmetry in the original templates' lenses (Figure 4 a,b), or to the resulting imprints being not deep enough (Figure 4 c,d, Figure S3). Both factors could make the determination of a focal point/image plane more complicated, and both factors could be in perspective overcome by fabricating alternative templates. Using variable templates adapted to more specific applications and use-cases is fundamentally enabled by the current demonstration.

REVIEWER COMMENTS

Reviewer #1 (Remarks to the Author):

In the rebuttal, the authors stress the novelty from their point of view. Although the effort is appreciated, I still believe the current work lacks novelty and significance. Many more articles, not just the initially addressed papers, report already from ZIF glass forming and hot-pressing. I agree with referee that Scientific Reports, or Communications Materials, is more suited for this work.

The added study of changes the reflective index is nice. However, it has been demonstrated for porous matters numerous times. So here too, the novelty is limited. Using such a porous glass to realize lenses with guest-dependent focal lengths would represent progress. However, this item is only outlined as a draft and not realized by the authors.

Reviewer #2 (Remarks to the Author):

After reading the whole manuscript, this revised version has addressed previous queries based on supplementary experiments and analysis of all the observed results. The paper quality has also been certainly improved after adopt the kind suggestions from others reviewers. On this occasion, I'm happy to suggest it is suitable for publication in Nature Communications.

Reviewer #3 (Remarks to the Author):

The authors invested substantial efforts in improving the manuscript in accordance with the reviewers' comments. The manuscript shows promising processing technique and applications of MOF glasses, e.g., in optics. Nevertheless, certain issues outlined below should be addressed prior to potential acceptance for publication in NC.

1. Introduction, on pages 3 and 4: The authors extensively describe previous research on MOF films. However, it's noted that some of these references are more aligned with coatings or in situ grow rather than the films prepared in the article. Hence, the introduction should be revised.
2. Pages 4 and 5: The authors dedicate considerable space to detailing the preparation of their samples, testing procedures, and methods. It's suggested that these details be relocated to the subsequent sections on results and methods instead of being included in the introduction. While the testing concepts and ideas can remain in the introduction with the combination with the past literatures, specific details should be deferred to later sections.
3. In line 147, the authors suggest that pore collapse would result in a decrease in viscosity. However, wouldn't a denser structure typically lead to an increase in the glass viscosity?
4. In line 182, the author relies solely on spectra to infer the ligand structural state of the material is unchanged before and after vitrification, which may lack sufficient evidence. It is recommended to include PDF testing to analyze changes in the short-range structure of the material before and after vitrification, thereby obtaining more convincing conclusions.
5. The work is of technological importance. However, to increase the scientific importance of their work, the authors should discuss the correlation between processing, glass structure optical properties. During imprinting, not only pores could collapse, but also the microstructure could undergo a large change. This is because ZIF glasses have very flexible, adaptive microstructure partly due to their high degree of short-range disorder (see Madsen et al, Science 2020, 367, 1473).
6. The term "agZIF-62 film" could be changed to "ZIF-62 glass film".

Reviewer #4 (Remarks to the Author):

The authors have answered all my questions and recommend publishing in Nature Communications.

Reviewer #1 (Remarks to the Author):

In the rebuttal, the authors stress the novelty from their point of view. Although the effort is appreciated, I still believe the current work lacks novelty and significance. Many more articles, not just the initially addressed papers, report already from ZIF glass forming and hot-pressing. I agree with referee that Scientific Reports, or Communications Materials, is more suited for this work.

The added study of changes the reflective index is nice. However, it has been demonstrated for porous matters numerous times. So here too, the novelty is limited. Using such a porous glass to realize lenses with guest-dependent focal lengths would represent progress. However, this item is only outlined as a draft and not realized by the authors.

We thank the reviewer for their time and consideration.

Reviewer #2 (Remarks to the Author):

After reading the whole manuscript, this revised version has addressed previous queries based on supplementary experiments and analysis of all the observed results. The paper quality has also been certainly improved after adopt the kind suggestions from others reviewers. On this occasion, I'm happy to suggest it is suitable for publication in Nature Communications.

We would like to thank the reviewer for their time and positive evaluation of our work.

Reviewer #3 (Remarks to the Author):

The authors invested substantial efforts in improving the manuscript in accordance with the reviewers' comments. The manuscript shows promising processing technique and applications of MOF glasses, e.g., in optics. Nevertheless, certain issues outlined below should be addressed prior to potential acceptance for publication in NC.

We thank the reviewer for their time and consideration. We agree that the manuscript has been improved during the revision process, thanks to the valuable suggestions of the four expert reviewers.

1. Introduction, on pages 3 and 4: The authors extensively describe previous research on MOF films. However, it's noted that some of these references are more aligned with coatings or in situ grow rather than the films prepared in the article. Hence, the introduction should be revised.

As noted by the reviewer, a certain part of the introduction is dedicated to crystalline MOF films, and not MOF glasses. As MOF glasses promise to combine some features of both worlds – MOFs and conventional glasses – we believe that mentioning the work that has been done to date in the field of crystalline MOFs is of importance, in particular, preserving the porosity of MOFs: the discussed film deposition reflects the state of the art, to which MOF glass processing is the potentially disrupting alternative. Therefore, MOF films with optical quality are discussed, and, along with the perspectives,

emphasis is given to the complicity of their fabrication (for instance, the sentence “*Fabricating dense polycrystalline MOF films with optical quality involves time-consuming, inefficient processing in a highly controlled environment.*”). Unlike glass, crystalline layers cannot be shaped and are indeed mostly formed *in situ*. In summary, we feel that our introductory comments on MOF films provide the reader with context and motivation towards delving into the world of MOF glasses.

2. Pages 4 and 5: The authors dedicate considerable space to detailing the preparation of their samples, testing procedures, and methods. It's suggested that these details be relocated to the subsequent sections on results and methods instead of being included in the introduction. While the testing concepts and ideas can remain in the introduction with the combination with the past literatures, specific details should be deferred to later sections.

We agree with the reviewer that some technical details can be removed from the introduction. The following sentences were removed from the introduction:

“For this purpose, we start with direct viscosity measurements closer to T_g (a_0 ZIF-62), as earlier data were available only at high temperatures or from extrapolation using the value of T_g and the calorimetric fragility index^[6].”

“Using imprinting tools generated by two-photon polymerization and sintering of silica precursors (Figure 1c), concave and convex optical components were subsequently fabricated from a_0 ZIF-62(Zn) using these viscosity data for input (Figure 1d,e).”

The following part was shortened and shifted to the chapter 2.2:

“As expected, we run into several effects that make handling of a_0 ZIF-62(Zn) challenging, such as shrinking of the glass melt due to collapsing porosity,^[9] which requires intricate process control in order to achieve large, crack-free objects.^[6,17,25] Another complication in the shaping of a_0 ZIF-62 lies in the hybrid organic-inorganic nature, which makes it essential to handle the melt under an inert atmosphere. For example, even minor traces of oxidizing gases can trigger partial linker decomposition and loss of transparency^[9]. We point out the crucial role of tackling these challenges in order to take hybrid glass applications from speculation to reality. At the same time, processing-dependent microporosity remains a unique feature of this class of materials, and in combination with its glassy nature offers broad opportunities for manufacturing responsive optical elements.”

3. In line 147, the authors suggest that pore collapse would result in a decrease in viscosity. However, wouldn't a denser structure typically lead to an increase in the glass viscosity?

In this case, the continuous process of pores collapsing during the measurement is considered in addition to viscous flow, not the equilibrium viscosity of the non-porous dense glass. This is why we denote the observed parameter “apparent viscosity” and, later, discuss the non-equilibrium nature of the TMA/penetration viscometry. Pore collapse while heating (addressed as “ongoing” in the manuscript) increases the rate of the ball’s sink-in into the glass and therefore influences the density value (addressed as “apparent”). While the reviewer is right in that a denser structure could intuitively lead to an increase in (equilibrium) viscosity, this is probably not always that clear.

4. In line 182, the author relies solely on spectra to infer the ligand structural state of the material

is unchanged before and after vitrification, which may lack sufficient evidence. It is recommended to include PDF testing to analyze changes in the short-range structure of the material before and after vitrification, thereby obtaining more convincing conclusions.

The reviewer is very right in that PDF has been used to characterize transient and permanent structural changes in MOF-derived glasses. As for our current report, our statement on the ligands remaining unchanged after melting is supported in the text not solely by the new spectra, but also by our previously published results (Nature Materials, 23, 262-270 (2024)), which is reflected in the same paragraph in the sentence “*This is in agreement with previous observations...*”. As for the suggestion to perform the PDF investigation, we think that this goes beyond the focus of the present study, given the extensive body of literature already reporting PDFs for this very material, in various scenarios of thermal treatment: there are many comprehensive studies including PDF investigations of ZIF-62 glass materials. In comparison to these, we would not expect significant changes or new insight from a straightforward before-after measurement. More advanced study on the precise influence of the preparation/treatment parameters on short-range structure appears exciting, but is very resource- and time-consuming and would certainly deserve a separate, structured and comprehensive study, perhaps for a more specialized audience.

5. The work is of technological importance. However, to increase the scientific importance of their work, the authors should discuss the correlation between processing, glass structure optical properties. During imprinting, not only pores could collapse, but also the microstructure could undergo a large change. This is because ZIF glasses have very flexible, adaptive microstructure partly due to their high degree of short-range disorder (see Madsen et al, Science 2020, 367, 1473.

The new paragraph has been added along with a few new references (48-49, including Madsen et al.):

As has been shown before, properties of ZIF-62 glass are highly dependent on each step of its production – from synthesis of the crystalline material^[6,9,48] to amorphization approach and further treatment^[9,25,49]. Not only porosity, but overall flexible and adaptive microstructure can undergo significant changes due to the high degree of short-range disorder^[48], influencing adsorption and optical behavior. Therefore, synthetic and any processing conditions must be thoroughly controlled – and then, in perspective, it will allow for more parameters and possibilities for precise tuning of application-oriented properties.

6. The term “agZIF-62 film” could be changed to “ZIF-62 glass film”.

The change has been applied. From:

“The second approach (Figure 1e, SI) used a post-processing technique of a_gZIF-62 films prepared beforehand.”

To:

“The second approach (Figure 1e, SI) used a post-processing technique of ZIF-62 glass films prepared beforehand.”

Reviewer #4 (Remarks to the Author):

The authors have answered all my questions and recommend publishing in Nature Communications.

We would like to thank the reviewer for their time and positive evaluation of our work.

REVIEWERS' COMMENTS

Reviewer #3 (Remarks to the Author):

The authors have addressed my comments and suggestions, and adequately improved manuscript. Therefore, I recommend accepting the manuscript in Nat Commun.